# Impact of cooking style and oil on semi-volatile and intermediate volatility organic compound emissions from Chinese domestic cooking

Kai Song[1,2], Song Guo[1,2,*], Yuanzheng Gong[1], Daqi Lv[1], Yuan Zhang[1,3], Zichao Wan[1], Tianyu Li[1], Wenfei Zhu[1], Hui Wang[1], Ying Yu[1], Rui Tan[1], Ruizhe Shen[1], Sihua Lu[1], Shuangde Li[4], Yunfa Chen[4], Min Hu[1,2]

[1] State Key Joint Laboratory of Environmental Simulation and Pollution Control, International Joint Laboratory for Regional Pollution Control, Ministry of Education (IJRC), College of Environmental Sciences and Engineering, *Beijing* 100871, China

[2] Collaborative Innovation Center of Atmospheric Environment and Equipment Technology, Nanjing University of Information Science & Technology, *Nanjing* 210044, China

[3] School of Earth Science and Engineering, Hebei University of Engineering, *Handan* 056038, China.

[4] State Key Laboratory of Multiphase Complex Systems, Institute of Process Engineering, Chinese Academy of Sciences, *Beijing* 100190, China

* **Correspondence:** Song Guo: songguo@pku.edu.cn

**Abstract:**

To elucidate the molecular chemical compositions, volatility-polarity distributions, as well as influencing factors of Chinese cooking emissions, a comprehensive cooking emission experiment was conducted. Volatile organic compounds (VOCs), intermediate volatility, and semi-volatile organic compounds (I/SVOCs) from cooking fumes were analyzed by a thermal desorption comprehensive two-dimensional gas chromatography coupled with quadrupole mass spectrometer (TD-GC×GC-qMS). Emissions from four typical Chinese dishes, i.e., fried chicken, Kung Pao chicken, pan-fried tofu, and stir-fried cabbage were investigated to illustrate the impact of cooking style and material. Fumes of chicken fried with corn, peanut, soybean, and sunflower oils were investigated to demonstrate the influence of cooking oil. A total of 201 chemicals were quantified. Kung Pao chicken emitted more pollutants than other dishes due to its rather intense cooking method. Aromatics and oxygenated compounds were extensively detected among meat-related cooking fumes, while a vegetable-related profile was observed in the emissions of stir-fried cabbage. Ozone formation potential (OFP) was dominated by chemicals in the VOC range. 10.2% - 32.0% of the SOA estimation could be explained by S/IVOCs. Pixel-based partial least squares-discriminant analysis (PLS-DA) and multiway principal component analysis (MPCA) were utilized for sample classification and component identification. The results indicated that the oil factor explained more variance of chemical compositions than the cooking style factor. MPCA results emphasize the importance of the unsaturated fatty acid-alkadienal-volatile products mechanism (oil autooxidation) accelerated by the cooking and heating procedure.

**Keywords:** Cooking emissions; Semi-volatile organic compounds; Intermediate volatility organic compounds; Cooking style; Oil

# 1   Introduction


Organics are key components of urban particles (Guo et al., 2014; Tang et al., 2018). Source
apportionment results indicated that vehicle exhaust is one of the important sources of gaseous and
particulate organics (Guo et al., 2012, 2020; Hu et al., 2015; Wang et al., 2021). However, the
importance of cooking emissions is rising due to the high impact on both primary precursor
emissions and secondary formation (Zhu et al., 2021). Cooking emitted organics are complex
mixtures covering a wide range of volatility, including volatile organic compounds (VOCs, organics
with effective saturation concentration higher than $10^6$ µg m$^{-3}$) (Bruns et al., 2016; Fullana et al.,
2004; Huang et al., 2011; Lu et al., 2021; Zhang et al., 2019), intermediate volatility organic
compounds (IVOCs, organics with effective saturation concentration in the range of $10^3$ -$10^6$ µg m$^{-3}$)
(Liu et al., 2018; Lu et al., 2021; Schauer et al., 2002), and semi-volatile organic compounds
(SVOCs, organics with effective saturation concentration in the range of $10^{-1}$ -$10^3$ µg m$^{-3}$) (Liu et al.,
2018; Lu et al., 2021; Ma et al., 2021; Schauer et al., 2002; Vicente et al., 2021; Yu et al., 2021).
Along with a large variety of volatility, these organics are also a large pool of complex components
of different polarities, such as alkanes with lower polarity (Gysel et al., 2018; Lin et al., 2010; Wang
et al., 2015), polycyclic aromatics with intermediate polarity (Chen et al., 2019; Kim et al., 2013;
Wei See et al., 2006), acids, ketones, and aldehydes with higher polarity (Alves et al., 2012; Gysel et
al., 2018; He et al., 2004; Peng et al., 2017). Such cooking-related organics are key pollutants
exhibiting health effects (Gligorovski et al., 2018; Huang et al., 2011; Zhao and Zhao, 2018) and air-
quality problems (Abdullahi et al., 2013; Zhao and Zhao, 2018). Although chemical compositions,
fingerprints, and influencing factors of cooking emissions have been investigated in some previous
studies (Alves et al., 2021; Klein et al., 2016a; Peng et al., 2017; Vicente et al., 2021), there are still
questions that remain uncertain. The first constraint is that resolving complex mixtures of cooking
emissions is rather tough. Most components in traditional gas chromatography-mass spectrometer
(GC-MS) chromatograms remain unresolved (Takhar et al., 2021; Zhao et al., 2014). It is of vital
importance to identify chemical compositions of unresolved complex mixtures (UCM) to better
understand their contributions to secondary organic aerosol (SOA). For instance, Huo et al
investigated the S/IVOC emissions from incomplete combustion utilizing GC×GC-MS. They found
that the previous bins-based method caused SOA underestimation with the ratio of 62.5 ± 25.2% to
80.9 ± 2.8% (Huo et al., 2021). Particle-phase SVOC organics from cooking emissions are widely
demonstrated yet few studies focus on gas-phase IVOC or SVOC organics. Meanwhile, current
studies mainly focus on a single kind or a series of homologs (aldehydes (Abdullahi et al., 2013;
Klein et al., 2016a; Peng et al., 2017), alkanes (Abdullahi et al., 2013), or acids (Abdullahi et al.,
2013; Takhar et al., 2021; Zeng et al., 2020)). In other words, currently, there are few comprehensive
source profiles of cooking emissions covering VOCs, IVOCs, and SVOCs (Schauer et al., 1999; Yu
et al., 2022).
The volatility-based method originated from the volatility-based set (VBS) is widely used to
demonstrate IVOC or SVOC emissions from different sources (Zhao et al., 2014, 2017), yet
chemical compositions from cooking emissions could not be demonstrated well only from the
volatility perspective. Large proportions of acids, esters, polycyclic aromatic hydrocarbons (PAHs),
and *n*-alkanes expand a wide range of polarity. A novel scheme combining volatility and polarity
should be developed to better identify source emission characteristics.
Besides, it is well-known that cooking emissions vary dramatically with cooking style,
ingredients, food, oil, and temperature (Amouei Torkmahalleh et al., 2017; Klein et al., 2016b; Liu et
al., 2018; Takhar et al., 2021; Zhao et al., 2007b). Cooking style and oil are typical influencing
factors dominating the compositions of cooking fume (Klein et al., 2016a; Takhar et al., 2021; Zhang
et al., 2019). Some studies demonstrated the emission patterns of cooking fumes emphasizing the
influence of different dishes or cooking methods (Chen et al., 2018; Wang et al., 2020), and several
studies clarified the importance of *n*-alkanes (Zhao et al., 2007a), polycyclic aromatic hydrocarbons
(PAHs) (Abdel-Shafy and Mansour, 2016; Abdullahi et al., 2013), aldehydes (Katragadda et al., 2010;
Peng et al., 2017), and acids (Pei et al., 2016; Zeng et al., 2020; Zhao et al., 2007a) from cooking
emissions using various kinds of oils. However, few comprehensive investigations have been
reported that speciated the dominant influencing factor under multiple conditions of cooking
procedures.
In this work, a thermal desorption comprehensive two-dimensional gas chromatography
coupled with quadrupole mass spectrometer (TD-GC×GC-qMS) is utilized to resolve and quantify

gaseous organic emissions from the molecular level. GC×GC has been proved to be a powerful technique to resolve UCM in previous studies (Cordero et al., 2018; Zhang et al., 2021a). A two-dimensional panel combining the volatility and polarity properties of chemicals is developed to better understand organic emissions. The ozone formation potential (OFP) and SOA formation from gaseous precursors were estimated. To elucidate the main influencing factor of cooking emissions, pixel-based partial least squares-discriminant analysis (PLS-DA) was utilized. The main chemical reactions of cooking emission were further inferred by pixel-based multiway principal component analysis (MPCA).

## 2    Experimental description

### 2.1    Sampling and quantification

Four typical Chinese dishes, i.e., fried chicken, Kung Pao chicken, pan-fried tofu, and stir-fried cabbage, were cooked in corn oil in the laboratory of the Institute of Process Engineering, Chinese Academy of Sciences. The detailed cooking procedures could be found in Table S1 and elsewhere (Zhang et al., 2021b). Meanwhile, four types of oil (i.e., soybean, corn, sunflower, and peanut oil ) were used for frying chicken to illustrate the influence of oil. These four oils were chosen for chicken-frying as they are commonly consumed in China (especially soybean oil) (Jamet and Chaumet, 2016) and other countries worldwide (Awogbemi et al., 2019).

Cooking fumes were sampled directly without dilution. After collecting particles on quartz filters, gas-phase organics were sampled by pre-conditioned Tenax TA tubes (Gerstel 6 mm 97 OD, 4.5 mm ID glass tube filled with ∼290 mg Tenax TA) with a flow of 0.5 L min$^{-1}$. The removal of particles on the quartz filter in front of the Tenax TA tubes affects the S/IVOC measurements, causing positive and negative artifacts. Some of the gaseous SVOCs could be lost to sorption onto filters, and some particle-phase SVOCs could evaporate off the filter. The emission pattern of the particulate organics diverged from gas-phase organics, and a small overlap of species is identified. Aromatics, aldehydes, and short-chain acids mainly occurred in the gas-phase. For instance, the detection of short-chain olefinic aldehydes in the gas-phase was 40 times that of the particle-phase aldehydes. The artifacts of particulates on gas-phase aromatics and oxygenated compounds could be less than 5%. A typical system blank chromatogram is displayed in Figure S1. A daily blank

sampling of the air in the kitchen ventilator was conducted before cooking and was subtracted in the quantification procedure. The sampling time in this work is 15 ~ 30 min (0.5 L min$^{-1}$). All samples were frozen at -20°C before analyzing. A Tenax TA breakthrough experiment was conducted by sampling two adsorbent tubes in series. We sampled the first tube (sample tube) and the second tube (backup tube) simultaneously with a sampling time of 24h. No breakthrough was observed after 24h sampling (Figure S2). The total intensity of cooking emission chromatograms ($3.05 \times 10^9 - 14.17 \times 10^9$) falls in the range of the sample tube ($9.84 \times 10^9$), which was much higher than the intensity of the backup tube ($2.12 \times 10^9$) and the blank tube ($1.33 \times 10^9$, Figure S1). After subtracting the volume of the blank tube, the volume of the backup tube is less than 10% of the sample tube, indicating the breakthrough effect of the Tenax TA tubes could be neglected.

A thermal desorption comprehensive two-dimensional gas chromatography coupled with quadrupole mass spectrometer (TD-GC×GC-qMS, GC-MS TQ8050, Shimadzu, Japan) was utilized for sample analysis with a desorption temperature of 280 °C. The modulation period was 6s. See more detail in Table S2. As the first and second columns of GC×GC were non-polar SH-Rxi-1ms (30 m × 0.25 mm × 0.25 μm) and mid-polar BPX50 (2.5 m × 0.1 mm × 0.1 μm), the 1$^{st}$ retention time of a chemical is related to its volatility while 2$^{nd}$ retention time is related to polarity (Nabi et al., 2014; Nabi and Arey, 2017; Zushi et al., 2016). The total chromatogram was cut into volatility bins (B8 to B31 with a decrease in volatility) following the pipeline of previous studies (Tang et al., 2021; Zhao et al., 2014, 2017, 2018), while it was cut into slices by an increase of 0.5 s in the second retention time (called 2D bins, from P1 to P12 with an increase of polarity). For instance, C12 lies in B12 (saturated vapor concentration ~ $10^6$ μg m$^{-3}$, IVOC range) and P2 bins (low polarity). Benzophenone lies in B16 (saturated vapor concentration ~ $10^5$ μg m$^{-3}$, IVOC range) and P6 bins (medium to high polarity). A two-dimensional panel was developed in this way to investigate the emission of contaminants from aspects of their volatility and polarity properties (Song et al., 2022).

326 chemicals were quantified (Table S3) while 201 contaminants were detected (Table S4) in cooking fumes covering a wide range of VOCs, IVOCs, and SVOCs, including 25 aromatics, 19 *n*-alkanes, 100 oxygenated compounds (containing 7 acids, 10 alcohols, 29 aldehydes, 24 esters, 5 ketones, and others), 3 PAHs, and 54 other chemicals. The 1D retention time shift of most chemicals

is within 0.5 min, while the 2D retention time shift of most chemicals is within 0.1s (Table S4),
which is much less than the length of 1D (~ 8 min) and 2D (0.5s) bins. Most of the $R^2$ of external
calibration curves was between 0.90 – 1 (Table S5). Chemicals without standards are semi-quantified
by surrogates from the same class or *n*-alkanes in the same 1D bins (Table S3). The uncertainties of
semi-quantification of surrogates from the same class or *n*-alkanes were 27% and 69% (Table S6).
The average emission rates (μg min⁻¹) of (semi-)quantified chemicals are listed in Table S4.

Quartz filters added with about 1 mL of edible oils were also thermally desorbed and analyzed

by TD-GC×GC-qMS. The total responses of blobs are normalized to 1 and the results were given by
percent response (%).

**2.2    Emission rate calculation, estimation of ozone and secondary organic aerosol (SOA) formation potential**

Emission rate (ER, μg min⁻¹) was calculated by the following equation, where $c$ is the blank

subtracted mass concentration (μg m⁻³) of the chemical quantified, and $Q$ is the mass flow of cooking
exhaust emissions (15 m³ min⁻¹).

$$ER = c \times Q \qquad (1)$$

Ozone formation potential (OFP, μg min⁻¹) was calculated by the following equation (Atkinson

and Arey, 2003),

$$OFP = \sum[HC_i] \times MIR_i \qquad (2)$$

Where $[HC_i]$ is the emission rate of precursor $i$ (μg min⁻¹) with maximum incremental reactivity

(MIR) of $MIR_i$. The MIR could be found in Table S3 and calculation procedures could be found
inside the FOQAT packages developed by Tianshu Chen (https://github.com/tianshu129/foqat).

SOA (μg min⁻¹) was estimated by the following equation, where $[HC_i]$ is the emission rate of

precursor $i$ (μg min⁻¹) with OH reaction rate of $k_{OH,i}$, (cm³·molecules⁻¹·s⁻¹) and SOA yield of $Y_i$
(Table S3). The SOA yields of precursors were from literature (Algrim and Ziemann, 2016, 2019;
Chan et al., 2009, 2010; Harvey and Petrucci, 2015; Li et al., 2016; Liu et al., 2018; Loza et al., 2014;
Matsunaga et al., 2009; McDonald et al., 2018; Shah et al., 2020; Tkacik et al., 2012; Wu et al., 2017)
or surrogates from *n*-alkanes in the same volatility bins (Zhao et al., 2014, 2017). The SOA yields
utilized in this work are under high NO$_x$ conditions which are underestimation of SOA due to the
lower yields compared to low $NO_x$ conditions. $[OH] \times \Delta t$ is the OH exposure and was set to be 14.4
$\times 10^{10}$ molecules·cm$^{-3}$·s (~ 1.1 days in OH concentration of $1.5 \times 10^6$ molecules·cm$^{-3}$) in order to keep
pace with our previous work (Zhang et al., 2021b; Zhu et al., 2021).

$$SOA = \sum [HC_i] \times (1 - e^{-k_{OH,i} \times [OH] \times \Delta t}) \times Y_i \ (3)$$

**2.3  Pixel-based analysis to demonstrate the main influencing factor of cooking emissions**

Pixel-based analysis was widely used as a dimension reduction tool for data interpretation
(Furbo et al., 2014). Pixel-based approaches have been proved to be powerful techniques for the
identification of atmospheric gaseous fingerprints (Song et al., 2022). In this work, pixel-based
partial least squares-discriminant analysis (PLS-DA) and multiway principal component analysis
(MPCA) were utilized for sample classification and key components identification, following the
pipeline of RGC×GC toolbox (Quiroz-Moreno et al., 2020). Chromatograms were imported from the
network common data form (netCDF). Smoothing, baseline correction, alignment, and
chromatogram unfolding were then conducted. MPCA was calculated inside the R language, while
PLS-DA was conducted by the interface of RGC×GC and mixOmics packages (González et al., 2012;
Lê Cao et al., 2009; Rohart et al., 2017). See more information about the data processing procedure
elsewhere (Quiroz-Moreno et al., 2020; Song et al., 2022).
PLS-DA is a supervised method for the classification of grouped data. The main influencing
factor could be apportioned if one separation result of PLS-DA is much better than the other. MPCA
composes matrix $X_{(i,j)}$ into score (S) and loading (L) matrices. Pixel-based MPCA could identify the
similarities by resolving chemicals from the positive loading chromatogram (Song et al., 2022).
All data processing was accomplished by GC Image® (GC×GC Software, 2.8r2, USA) and R
4.1.0 (Chen, 2021; Patil, 2021; R Core Team, 2020).

**3  Results and discussions**

**3.1  Molecular compositions of S/IVOCs, OFP, and SOA estimation from different dish fumes**

Typical chromatograms of four dish emissions are displayed in Figure S3. Chemicals identified
are colored in groups in Figure 1. The total mass concentrations of four dishes are displayed in
Figure 2. The emission rate of Kung Pao chicken was the highest (6918 ± 5924 μg min$^{-1}$), followed
by fried chicken (4827 ± 3308 μg min$^{-1}$), pan-fried tofu (3854 ± 3809 μg min$^{-1}$), and stir-fried
cabbage (697 $\pm$ 548 $\mu g\ min^{-1}$). Stir-frying procedures of Kung Pao chicken were rather intense,
followed by deep-frying chicken. Research has revealed that VOC emissions from quick- and stir-
frying or deep-frying cooking methods are much higher (Chen et al., 2018; Ciccone et al., 2020;
Kabir and Kim, 2011; Lu et al., 2021).
The compositions of the gaseous emissions are exhibited in Figure S4. Aromatics contributed
59.1%, 23.6%, 8.1%, and 11.8% of the total mass concentration of Kung Pao chicken, fried chicken,
pan-fried tofu, and stir-fried cabbage, while oxygenated compounds accounted for 17.1%, 53.7%,
76.9%, and 25.0% of the total concentration, respectively. The compositions of organic in this study
diverged from proton transfer reaction mass spectrometer (PTR-MS) measurements (Klein et al.,
2016a; Liu et al., 2018), in which aldehydes dominated the emission profiles (~ 60%). The
proportion of aromatics was also different from online Vocus-PTR-ToF measurements in a recent
study (Yu et al., 2022). However, the contribution of aromatics was close to a recent study conducted
at Chinese restaurants using GC-MS analysis (Huang et al., 2020). The different instruments
resulting in different VOC detection ranges could be the explanation for the different patterns. GC$\times$
GC-MS is powerful in resolving complex mixtures with carbon numbers of more than 6. The
structural chromatograms and detailed mass spectrum information provide a convincing result in
chemical identification (An et al., 2021). In contrast, PTR-MS could detect much more short-chain
alkenes and aldehydes with carbon numbers less than 4. However, the isomers of PTR-MS could not
be distinguished. Alkanes and some long-chain compounds could not be detected by PTR-MS. For
instance, the maximum carbon number of pollutants in Yu et al is 16 ($C_{16}H_{26}$) (Yu et al., 2022) while
the maximum carbon number of pollutants detected in this work is 30 ($C_{30}H_{62}$). $C_2H_6O$, $C_4H_8$,
$C_4H_8O2$, and $C_5H_8$ were the top species measured by Vocus-PTR-ToF (Yu et al., 2022), which is out
of range of our measurement. Compositions of organic emissions diverged significantly and showed
a great influence pattern of cooking styles (Wang et al., 2020). Dishes cooked by intense cooking
methods, like stir-frying, released more aromatics. Despite this, researches have indicated that the
emission patterns of different cooking styles are heavily driven by the thin or thick layer of oil (oil
amount), oil temperature, evaporation of water during cooking, and chemical reactions, such as
starch gelatinization, and protein denaturation (Atamaleki et al., 2021; Zhang et al., 2020). As for

chemical species, toluene, hexanoic acid, and pentanoic acid were extensively detected among meat-related cooking fumes, which were among the top 5 species and accounted for more than half of the total emission rate. A vegetable-related pattern was observed in the emissions of stir-fried cabbage. Alkanes (C10 – C12), alcohols (linalool, butanol), and pinenes (beta-pinene) were the dominant chemical classes. As much as 26.3% and 26.1% of the total organics of stir-fried cabbage emission were alkanes and alkenes (especially pinenes). The high plant wax content (Zhao et al., 2007a) in this dish dramatically influenced the composition of the fume.

Although the profiles of compositions diverged from dish to dish, their volatility-polarity patterns remained similar, showing a consistent pattern with a recent study (Yu et al., 2022). The volatility-polarity distributions of the gaseous emissions are displayed in Figure 3. VOCs (B11 and before, saturated vapor concentration $> 10^6$ µg m$^{-3}$) with low polarity (P1 – P4) dominated the emissions of gas-phase contaminants. Chemicals in the VOC range accounted for 88.7%, 95.6%, 85.2%, and 81.4% of the total emission rates of fried chicken, Kung Pao chicken, pan-fried tofu, and stir-fried cabbage emissions, while S/IVOCs accounted for 11.3%, 4.4%, 14.8%, and 18.2%, respectively. However, considering the chemical compositions in each volatility bin, the emission patterns are quite distinct (Figure S5). Oxygenated compounds were widely detected before B13 (VOC-IVOC range) in emissions of fried chicken and pan-fried tofu, while aromatics were extensively detected in the B8 range of Kung Pao chicken fumes. Alkanes and alkenes in the B10 range dominated the emissions of stir-fried cabbage. From the discussion above, the volatility distribution of cooking emissions obtained from the one-dimensional GC-MS analysis faces large uncertainty in SOA estimation if the polarity is not taken into account. Meanwhile, the volatility-polarity distribution should be equipped with detailed chemical parameters in each bin to precisely estimate SOA.

The total emission rates, compositions, and volatility-polarity distributions of OFP and SOA estimation by gaseous precursors are displayed in Figure 2, Figure S4, and Figure 3, respectively. The total OFP and SOA estimation are consistent with the emission rate, as Kung Pao chicken emitted the most pollutants and produced the most ozone formation (21125 ± 19447 µg min$^{-1}$) and SOA formation (584 ± 482 µg min$^{-1}$). Pan-fried tofu emitted a little bit less than fried chicken, yet

produced more SOA estimation due to a large proportion of short-chain acids (hexanoic acid) (Alves

and Pio, 2005; Forstner et al., 1997; Kamens et al., 1999). Short-chain acids are likely derived from

scission reactions of allylic hydroperoxides originating from unsaturated fatty acids (Chow, 2007;

Goicoechea and Guillén, 2014). Although chemicals in the VOC range dominated ozone and SOA

formation, an increase in ozone formation contribution and a decrease in SOA formation contribution

compared with the mass proportion of VOCs in EFs were observed. VOCs contributed 90.3% - 99.8%

of the ozone estimation, and 68.0% - 89.8% of the total SOA estimation, compared with 81.4% -

95.6% in EFs. S/IVOCs explained 10.2% - 32.0% of the SOA estimation. Aromatics (toluene) and

alkenes (heptene) were dominant ozone formation precursors in meat-relating dishes (fried chicken,

Kung Pao chicken, and pan-fried tofu), while alcohols (butanol and linalool) were predominant for

stir-fried cabbage (Atamaleki et al., 2021). Acids (hexanoic acid), aromatics (toluene), alkenes

(pinenes), and alkanes were important SOA precursors. We also want to emphasize that there are

large uncertainties in SOA estimation. Yu et al measured gas-phase VOC, IVOC, and SVOC

precursors by Vocus-PTR-ToF and compared the results with SOA measured from the aerosol mass

spectrometer (AMS). 19 ~ 55% of the SOA could be explained. Among them, the SOA estimation

from precursors emitted from Kung Pao chicken is the largest even though the SOA mass is the

lowest among the four dished (Yu et al., 2022). The SOA estimation in this work is also the largest

regarding Kung Pao chicken emissions. Aromatics and alkenes in Kung Pao chicken fumes

contributed 63.6% of the SOA estimation, and the top SOA contributor in Yu et al. were

sesquiterpenes and aromatics, showing a consistent pattern between these two studies. It should be

noticed that more than 45% of the SOA could not be explained (Yu et al., 2022) and more

investigations should be carried on to further identify the emission and evolution of cooking fumes in

the atmosphere.

**3.2  Molecular compositions of S/IVOCs, OFP, and SOA estimation from fried chicken fumes**
**using four types of oils**

Typical chromatograms of fried chicken emissions cooked with corn, peanut, soybean, and

sunflower oils are displayed in Figure S6. Chemicals identified are colored in groups in Figure S7.

Total chemical emission rates were 4827 ± 3308 μg min$^{-1}$, 3423 ± 988 μg min$^{-1}$, 3625 ± 1834 μg min$^{-}$

[1], and 2268 µg min$^{-1}$ (n = 1) for chicken fried with corn, peanut, soybean, and sunflower oils,
respectively (Figure 4). Chicken fried with corn oil emitted the most abundant gaseous contaminants.
The emission patterns in this work diverged from heated oil fumes (Liu et al., 2018) as in their work
heated sunflower oil and peanut oil emitted more organics. Compositions and volatility-polarity
distributions of contaminants are displayed in Figure S8 and Figure S9, respectively. Aromatic
contributed 23.6%, 20.1%, 50.5%, and 19.8% of the total ERs of fried chicken fumes cooked with
corn, peanut, soybean, and sunflower, oils, respectively. Fried chicken fumes cooked with soybean
oil were especially abundant in toluene (rank 1$^{st}$). In the TD-GC×GC-MS analysis of soybean oil
(Figure S10), unsaturated fatty acids (linoleic acid) contributed 31.5% of the total percent response
(50.5% aromatics), compared to 10.1% of the total response in corn oil (15.5% aromatics). As a
result, the aromatic concentrations and compositions of the fried chicken fumes diverged according
to the content of unsaturated fatty acids in the oil (Chow, 2007; Zhang et al., 2019). Butanol was the
most abundant chemical when peanut and sunflower oils were used for frying. A previous study
indicated that benzene, toluene, and ethylbenzene were the three dominant aromatics in kitchens
(Huang et al., 2011; Yi et al., 2019). Monocyclic aromatics are formed from linoleic and linolenic
acyl groups in the oil (Atamaleki et al., 2021; Uriarte and Guillén, 2010). The decomposition of
linoleic and linolenic acid forms alkadienals and then form aromatics once lose $H_2O$ (Atamaleki et
al., 2021; Zhang et al., 2019). According to previous studies, soybean oil contains more unsaturated
fatty acids, especially linolenic acid (Kostik et al., 2013; Ryan et al., 2008). Oxygenated compounds
were extensively detected, which accounted for 53.7%, 33.1%, 24.7%, and 35.0% of the total ERs
(Figure S8). Short-chain acids and aldehydes were the most abundant oxygenated compounds and
were dominated by hexanoic acid, hexanal, and nonanal. Despite acids and aldehydes, alcohols
(butanol, octenal) were heavily detected in the fume of corn oil-fried chicken, which was also
supported by another study (Liu et al., 2018; Reyes-Villegas et al., 2018). The short-chain
contaminants were fundamentally formed by hydroperoxide decomposition (originated from oleate
and linoleate in the oil) through homolytic scission or homolytic $\beta$-scission reactions (Chow, 2007;
Goicoechea and Guillén, 2014) and quickly evaporated from the oil. Either aromatics or oxygenated
compounds detected in the gas phase showed high sensitivity to oil compositions, especially
potentially influenced by oleic and linoleic compounds.

Although pollutants were dominated by aromatics, alkanes, and oxygenated compounds with

volatility bins of B9 to B12 (VOC-IVOC range, saturated vapor concentration $> 10^6$ μg m$^{-3}$) and
polarity bins of P1 to P5 (low to medium polarity), significant diversities of volatility-polarity
distributions were observed (Figure S9). The chemical compositions in each volatility bin were also
distinct (Figure S11). IVOCs accounted for as much as 22.8% and 23.7% of the total ERs when
peanut and sunflower oils were utilized for frying (Kostik et al., 2013; Ryan et al., 2008). The peanut
oil was much more abundant in oleic acid (41.5%), while the proportion of linoleic acid in sunflower
is 36.6% (Figure S10). The proportion of unsaturated acids in peanut and sunflower oils is higher
than that of other oils.

Chicken fried in soybean oil produced the highest OFP (10134 ± 5958 μg min$^{-1}$) while chicken

fried in corn oil resulted in the most SOA estimation (426 ± 270 μg min$^{-1}$). Aromatics were
predominant in ozone formation, while oxygenated compounds, alkenes, alkanes, and aromatics
were important SOA precursors. S/IVOCs were non-negligible SOA precursors because they
contributed as much as 22.0%, 28.2%, 24.0%, and 29.7% of the SOA estimation. Without S/IVOCs,
a large proportion of SOA would be underestimated. Our work illustrated the importance of the
measurement of S/IVOC precursors which was absent in previous studies (Liu et al., 2018; Zhang et
al., 2021b). Despite the importance of aldehydes revealed in previous studies (Klein et al., 2016a;
Liu et al., 2018), our results demonstrated that alkanes, pinenes, and short-chain acids are also key
precursors in cooking SOA production (Huang et al., 2020).
**3.3    Elucidating the influencing factor and inferring in-oil reactions of cooking emissions**

From the discussion above, cooking style and oil could influence emissions dramatically. But

we still wonder what is the main predominant factor shaping the profile of cooking emission. In
other words, we want to learn whether the cooking styles affect cooking patterns more. A pixel-based
partial least squares-discriminant analysis (PLS-DA) was utilized to investigate the key factor. The
results are displayed in Figure 5. PLS-DA is a supervised classification method requiring the data
pre-grouping. The separation results of the PLS-DA indicate the crucial pattern behind the
classification. When oil was set as the grouping variable, the separation was much better than setting

the dish as the grouping variable (Figure 5 (a) and (b)). The separation results demonstrated that the oil used during the cooking procedure is much more crucial in shaping the emission profiles than the cooking style. The variance of cooking fumes could be largely explained by the different oil utilized.

Plenty of physical and chemical reactions occur during the cooking procedure (Chow, 2007; Goicoechea and Guillén, 2014). To demonstrate the direct effect of oil on cooking emissions, PLS-DA and MPCA analyses were utilized. The PLS-DA result showed that cooking emissions diverged from oils (Figure 5 (c)), indicating that the physical reactions (evaporation of edible oils) were not the main reactions during the cooking procedure. MPCA results showed the chromatogram similarities (positive loading) of oils and emissions (Figure 5(d)). Fatty acids (palmitic acid, oleic acid, and linoleic acid), decanal, and decadienals were the key fingerprints. The pattern is linked to the autooxidation procedure of oil. Oil autooxidation is a three-step free radical process: initiation, propagation, and termination (Atamaleki et al., 2021; Uriarte and Guillén, 2010; Yi et al., 2019). The key initiation step is the formation of lipid radical (R•) from unsaturated fatty acid (RH). R• then reacts with $O_2$ to form peroxyl radical (ROO•) and then form hydroperoxides (ROOH). Another RH changes to R• in this propagation process. During the termination process, the decomposition of ROOH forms monomeric (keto-, hydroxy-, and epoxy- derivatives), polymeric (RR, ROR, ROOR), and volatile compounds (short-chain acids, aldehydes, alcohols, ketones). In more detail, the oxidation of unsaturated fatty acids (such as linoleic acid) in oil leads to the production of alkadienals (such as (*E*, *E*)-2,4-decadienal) which form aromatics (butylbenzene) by losing $H_2O$ (Atamaleki et al., 2021; Zhang et al., 2019). This is consistent with the analysis of edible oils in this work. Corn oil contained a less amount of unsaturated fatty acids (Figure S10), and the emission of aromatics cooked with coil oil was the lowest among the 4 types of oils used. The emission pattern is in line with previous studies (Atamaleki et al., 2021). The short-chain aldehydes and acids are derived from scission reactions of allylic hydroperoxides originated from unsaturated fatty acids (Chow, 2007; Goicoechea and Guillén, 2014), while the dehydration reaction of alkenals forms furanones (Zhang et al., 2019). Aldehydes, acids, and furanones are regarded as potential tracers of cooking emissions (Klein et al., 2016a; Wang et al., 2020; Zeng et al., 2020) and were widely detected in this work. These highly volatile contaminants escape from oil immediately and lead to an

accumulation of oxygenated compounds in the gas phase. Figure S12 shows the inferred reactions
originating from linoleic acid and oleic acid. The significant correlations ($p < 0.1$) between key
components (Figure S13) further support the chemical reactions demonstrated in Figure S12. The
key chemicals elucidated by the MPCA analysis (Figure 5 (d)) illustrated that the cooking emissions
are largely driven by the autooxidation of oil, which is accelerated during the heating and cooking
procedures (Atamaleki et al., 2021; Uriarte and Guillén, 2010; Yi et al., 2019; Zhang et al., 2019).
**4   Atmospheric Implications**

In this work, gaseous VOCs, IVOCs, and SVOCs from cooking fumes are quantified in detail.

The influence of cooking style and oil is taken into account in this work. S/IVOC species are key
components as they contributed 10.2% - 32.0% of the total SOA estimation. Previous works might
underestimate the importance of cooking fumes to SOA formation because only a series of IVOC
homologs were quantified (Liu et al., 2018). For instance, aldehydes only accounted for 0.7% -10.1%
of the total SOA estimation. If only aldehydes are taken into consideration, SOA will be
underestimated 9.9 ~ 139 times. We still need to stress that although GC×GC is utilized, UCMs still
occur sharing a proportion of 5% of the total response in this work. Acids and aldehydes tail in the
second column and cause uncertainties in the quantification procedure. Meanwhile, TD-GC×GC-MS
does not comprehensively measure all compounds. Acids can decompose during thermal desorption
if no derivatization was performed. Meanwhile, the decomposition of SVOC compounds could
produce small molecules in the VOC or IVOC range. The TD process could introduce
underestimation for SVOC compounds while causing overestimations of VOC and IVOC species.
Highly polar compounds do not elute from the GC column. This may lead to biases in estimating
volatility and polarity distributions. Comparisons between GC×GC and chemical ionization mass
spectrometers (CIMS) should be further implemented to give a full glimpse of cooking organic
compounds.

We also first proposed a novel two-dimensional panel elucidating the physiochemical properties

of contaminants from the perspectives of their volatilities and polarities. This novel scheme is
appropriate to demonstrate the complicated evolution of contaminants clearly and provide new
insight into the previously 1D-bins method. The volatility-polarity panel inherited the spirit of the

two-dimensional volatility-based set (2D-VBS) (Donahue et al., 2011, 2012) and would be further implemented in the analysis of complex ambient or source samples along with the powerful separating capacity of GC×GC. We would like to emphasize the importance of combining the volatility-polarity distribution with detailed chemical information for a precise estimation of SOA.

We also provide powerful tools in speciating the main driving factor and inferring chemical reactions in rather complicated systems. The pixel-based PLS-DA and MPCA analysis greatly enhance our learning of complex chromatograms and provide us with new insight into the dimension reduction processes. The analyzing scheme could benefit those analysts with less experience in GC×GC data processing.

Our results demonstrated that both cooking styles (dish) and oils influence the cooking emissions. Kung Pao chicken emitted more pollutants than other dishes due to its rather intense cooking method. Cooking materials could also influence the compositions of fumes as well. Aromatics and oxygenated compounds were extensively detected among meat-related cooking fumes, while a vegetable-related pattern was observed in the emissions of stir-fried cabbage. As much as 22.2% and 29.5% of the total organics of stir-fried cabbage emission were alkanes and alkenes (especially pinenes). On the other hand, oils greatly influence the composition and volatility-polarity distribution of pollutants. Chicken fried with corn oil emitted the most abundant contaminants. However, the ozone formation from soybean-oil fried chicken fumes was much higher. Considering the high consumption proportion of soybean oil (~ 44% in volume of oil usage) in China (Jamet and Chaumet, 2016), the influence of using soybean cooking oil on ozone formation might be underestimated. The MPCA results also indicate that the heating and cooking procedure greatly enhances the autooxidation of oil. MPCA results emphasize the importance of the unsaturated fatty acid-alkadienal-volatile product mechanism. More studies need to be carried on to elucidate the key chemical reactions between the food and oil.

## Acknowledgment

The work was funded by National Natural Science Foundation of China (No. 41977179, 91844301), the special fund of State Key Joint Labotatory of Environment Simulation and Pollution

Control (No. 22Y01SSPCP), the Open Research Fund of State Key Laboratory of Multi-phase
Complex Systems (No. MPCS-2021-D-12). We greatly thank Mengxue Tong for the sample
collection.

**Credit Author Statement:**
Kai Song, Yuanzheng Gong, and Daqi Lv conducted the experiments.
Kai Song and Yuanzheng Gong analyzed the data.
Kai Song, Song Guo, Yuanzheng Gong, Daqi Lv, Yuan Zhang, Zichao Wan, Tianyu Li, Wenfei Zhu,
Hui Wang, Ying Yu, Rui Tan, Ruizhe Shen, Sihua Lu, Shuangde Li, Yunfa Chen, and Min Hu
discussed the scientific results and review the paper.
Kai Song and Song Guo wrote the paper.

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

**Figure 1.** Chemical identified from fried chicken (a), Kung Pao chicken (b), Pan-fried tofu (c), and stir-fried cabbage (d) emissions. Column and Tenax TA bleeding after 75 min in 1$^{st}$ retention time are excluded from qualification, quantification, and 2D binning processes. Blobs are colored by chemical groups.

**Figure 2.** Emission rate (ER), ozone formation potential (OFP), and secondary organic aerosol (SOA) estimation from emissions of fried chicken, Kung Pao chicken, pan-fried tofu, and stir-fried cabbage. The unit of the $y$-axis is μg min$^{-1}$.

**Figure 3.** Volatility-polarity panels of gaseous chemical emissions from fried chicken, Kung Pao chicken, pan-fried tofu, and stir-fried cabbage fumes, and ozone formation potential (OFP), and secondary organic aerosol (SOA) estimation from gas-phase precursors. VOCs (blue color in $x$-axis), IVOCs (orange color in $x$-axis), and SVOCs (red color in $x$-axis) are displayed in volatility bins (a decrease of volatility from B9 to B31) along with their polarity (an increase from P1 to P10 in $y$-axis). The emission rate (ER) unit is μg min$^{-1}$.

**Figure 4.** Emission rate (ER), ozone formation potential (OFP), and secondary organic aerosol (SOA) estimation from emissions of fried chicken cooked with corn, peanut, soybean, and sunflower oils. The unit of the $y$-axis is μg min$^{-1}$.

**Figure 5.** PLS-DA classification results in setting the cooking style (a) or oil (b) as grouping variables. When oil was set as the grouping variable, the separation of groups was much better than setting the dish as the grouping variable. The PLS-DA comparison result of cooking emissions and oils is displayed in (c), indicating that the cooking fume is not just the evaporation of oil itself. Positive loadings of oil and cooking fume chromatograms (d) demonstrated the key components contributing to the similarities of samples.

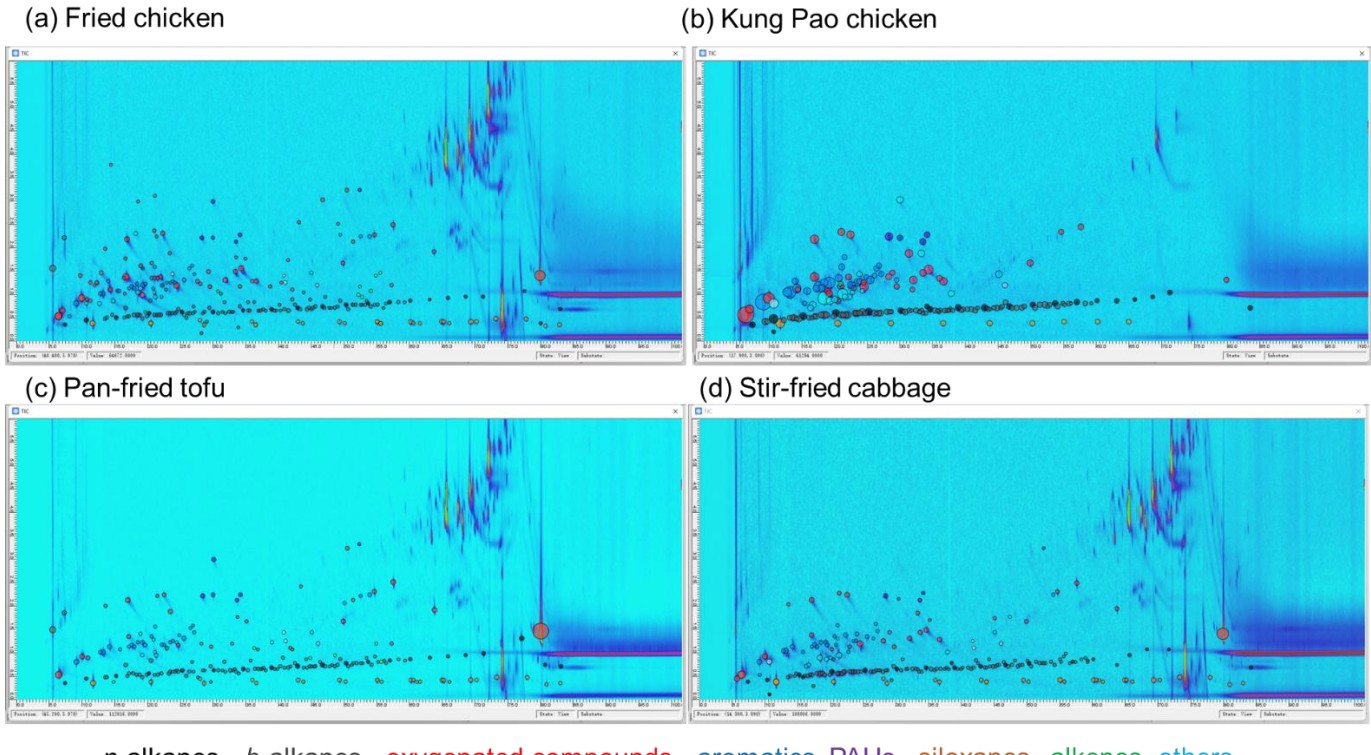

(a) Fried chicken (b) Kung Pao chicken

(c) Pan-fried tofu (d) Stir-fried cabbage

*n*-alkanes  *b*-alkanes  oxygenated compounds  aromatics  PAHs  siloxanes  alkenes  others

**Figure 1.** Chemical identified from fried chicken (a), Kung Pao chicken (b), Pan-fried tofu (c), and stir-fried cabbage (d) emissions. Column and Tenax TA bleeding after 75 min in 1$^{st}$ retention time are excluded from qualification, quantification, and 2D binning processes. Blobs are colored by chemical groups.

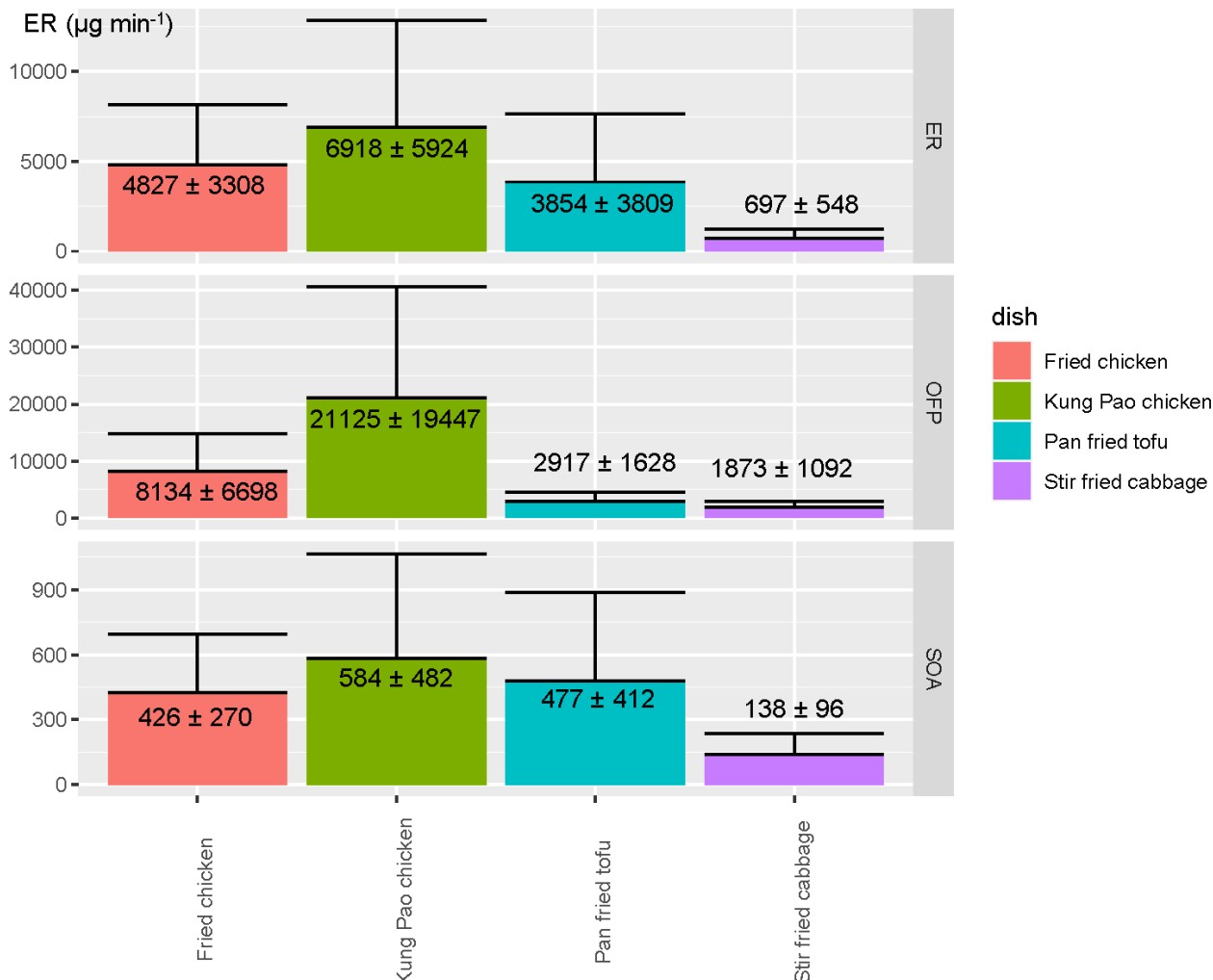


**Figure 2.** Emission rate (ER), ozone formation potential (OFP), and secondary organic aerosol (SOA)
estimation from emissions of fried chicken, Kung Pao chicken, pan-fried tofu, and stir-fried cabbage.
The unit of the *y*-axis is μg min$^{-1}$.

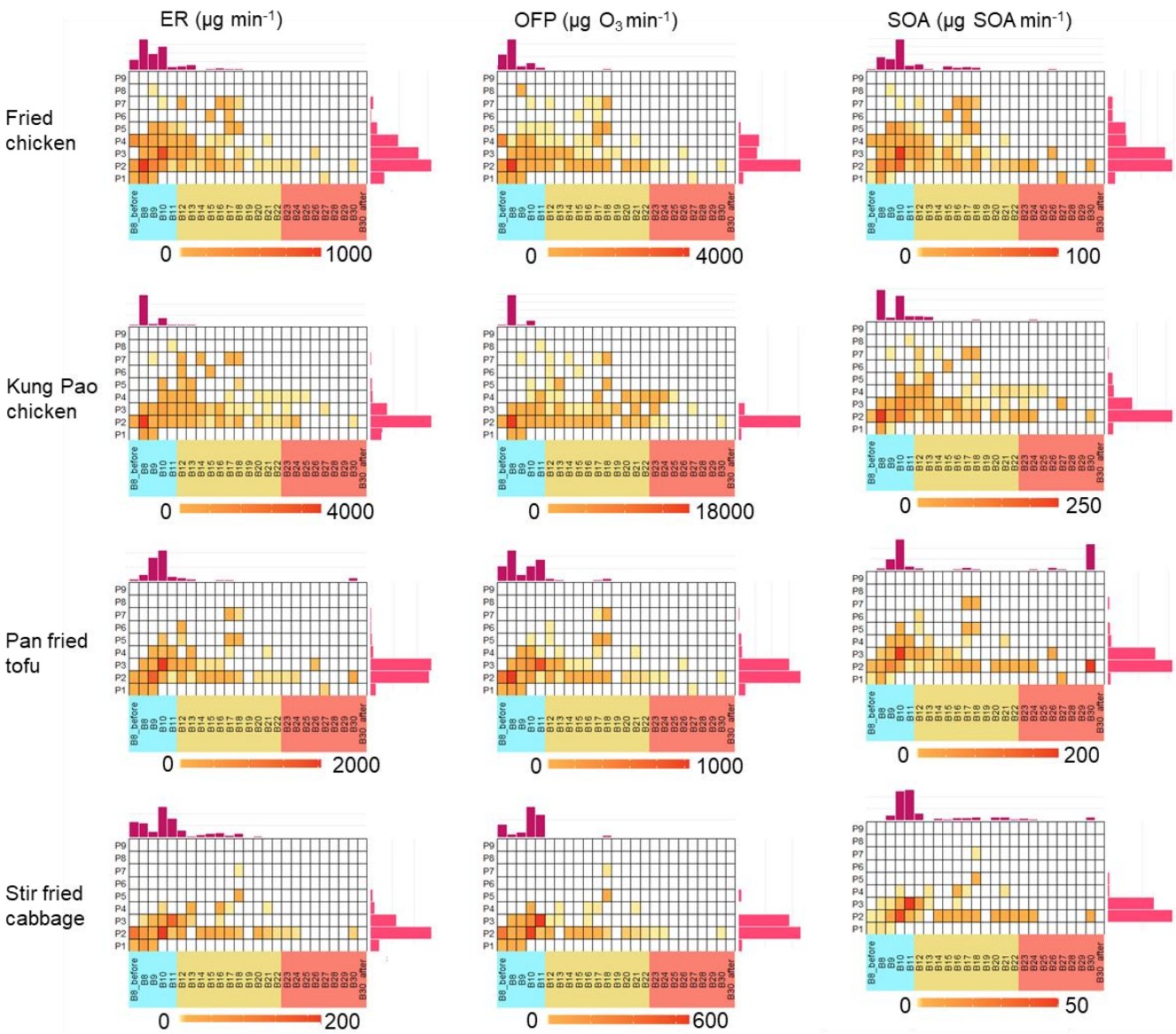

**Figure 3.** Volatility-polarity panels of gaseous chemical emissions from fried chicken, Kung Pao chicken, pan-fried tofu, and stir-fried cabbage fumes, ozone formation potential (OFP), and secondary organic aerosol (SOA) estimation from gas-phase precursors. VOCs (blue color in *x*-axis), IVOCs (orange color in *x*-axis), and SVOCs (red color in *x*-axis) are displayed in volatility bins (a decrease of volatility from B9 to B31) along with their polarity (an increase from P1 to P10 in *y*-axis). The emission rate (ER) unit is μg min$^{-1}$.

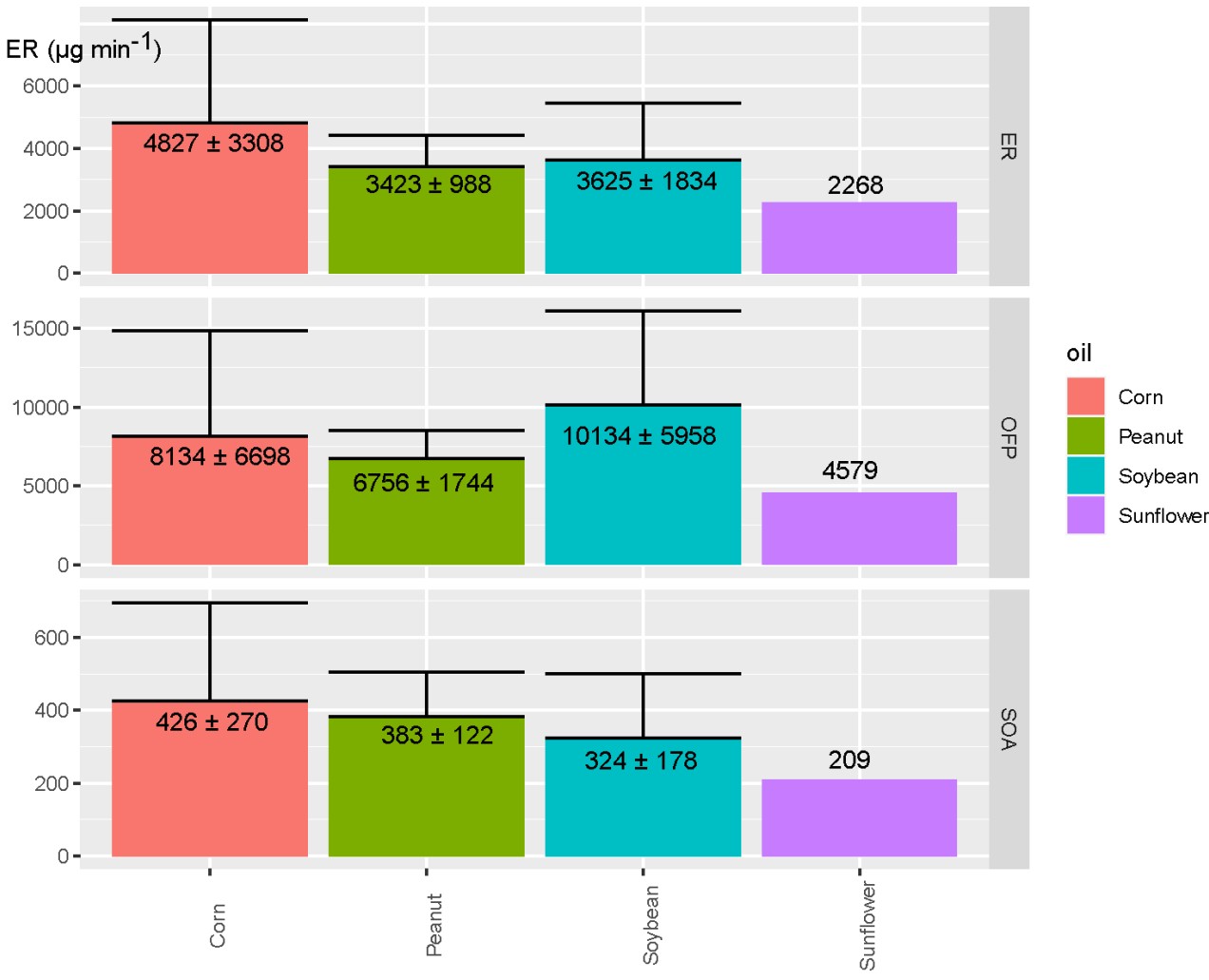

Figure 4. Emission rate (ER), ozone formation potential (OFP), and secondary organic aerosol (SOA) estimation from emissions of fried chicken cooked with corn, peanut, soybean, and sunflower oils. The unit of the *y*-axis is μg min⁻¹.

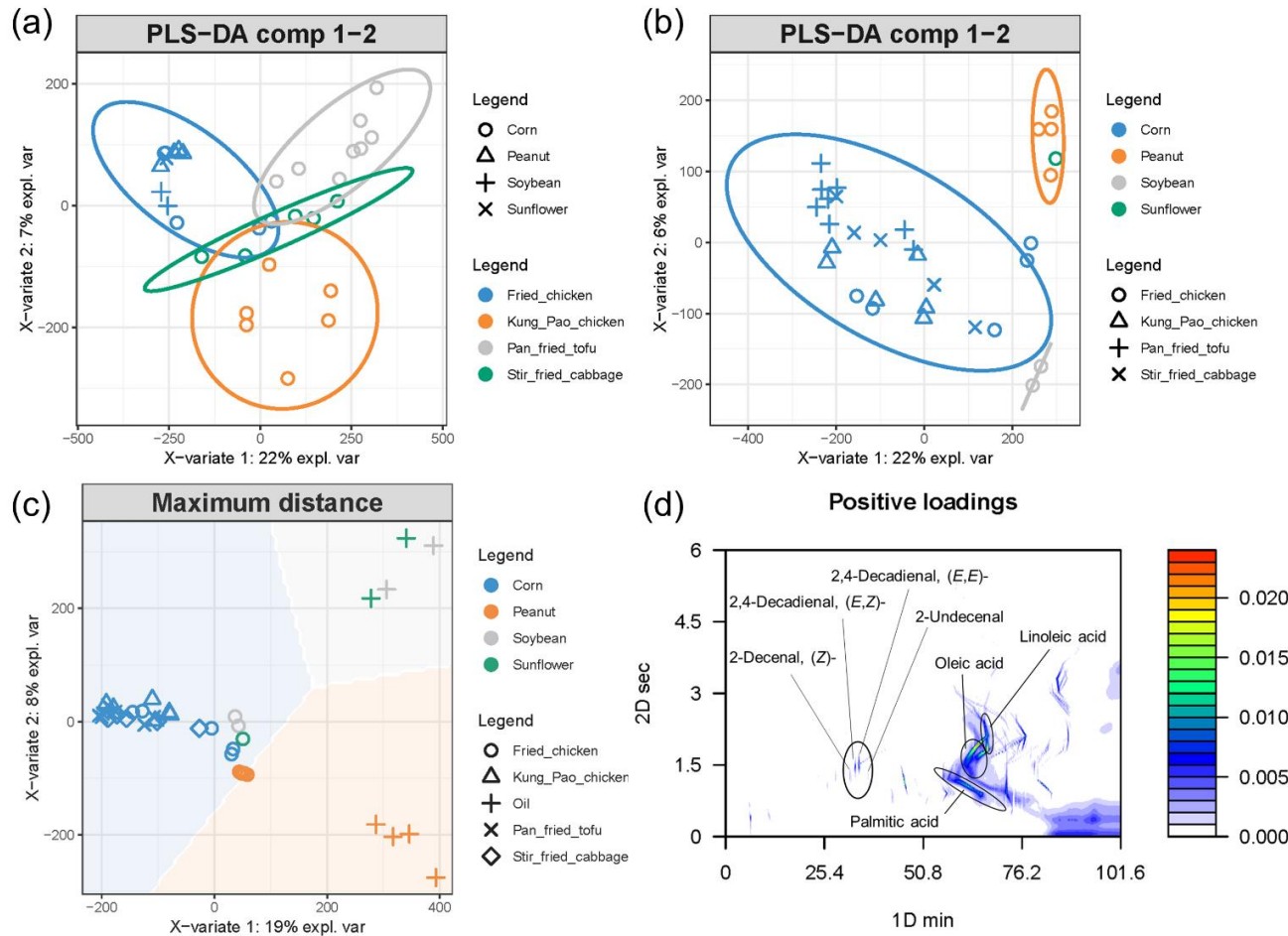


**Figure 5.** PLS-DA classification results in setting the cooking style (a) or oil (b) as grouping
variables. When oil was set as the grouping variable, the separation of groups was much better than
setting the dish as the grouping variable. The PLS-DA comparison result of cooking emissions and
oils is displayed in (c), indicating that the cooking fume is not just the evaporation of oil itself.
Positive loadings of oil and cooking fume chromatograms (d) demonstrated the key components
contributing to the similarities of samples. The color bar in (d) is the positive loading of pixels.