# Peer review of "Impact of cooking style and oil on semi-volatile and intermediate"

_Atmospheric Chemistry and Physics, 2022_

## Author Comment (AC1)

We thank the reviewers for their careful review of the manuscript. The comments greatly improved our manuscript. We revised our manuscript according to the reviewers' comments and suggestions. Overall, we have changed the mass concentration ($\mu g\ m^{-3}$) to the emission rate ($\mu g\ min^{-1}$) to avoid the influence of cooking time and sampling time according to the comments of the referees. We add more details to the volatility distributions of cooking emissions. We also added more comparisons with different studies. Following are our responses to the comments.

**Response to referee #1:**

In this manuscript, the authors studied the gaseous emissions from Chinese domestic cooking and the impact of cooking style and oil used on the emission profiles. They performed cooking experiments at a laboratory facility and measured the detailed composition of gaseous compounds using multidimensional GC-MS. They observed that the oil type played the most important role in determining the volatility and polarity distribution of compounds, while the type of food cooked and cooking style influenced the detailed composition, but was less of a factor in determining overall volatility and polarity. They also highlighted the role of IVOCs and SVOCs, which are not as well measured in previous studies but can add 10-30% to estimated SOA formation. All of these observations are important for understanding food cooking emissions as a source of reactive organic compounds in the urban atmosphere. The experiments are well-designed and the results are thoroughly interpreted and explained. The manuscript is often difficult to understand so I would recommend major revisions, mostly for the sake of improving the clarity of the manuscript. Otherwise the technical content is suitable for publication in ACP.

We greatly thank the reviewer for his careful review of the manuscript. Following are our point-to-point responses to the comments.

Major comments:

All of the emissions are reported in air concentrations (ug per m3 of air sampled). These numbers would depend on air flow rate through the cooking apparatus, which may vary between experiments. Have the authors verified that the flow rate is consistent between experiments? Also, the VOCs are collected in integrated samples, so the duration of sampling would matter too, which may vary depending on cooking times. I looked at the paper referenced (Zhang et al, ES&TL 2021) and it seems like cooking times are ~60 min and the sampling times are ~90min, but the flow rates are not known.

Even if the air flow rate is controlled, it is difficult to compare these numbers to other experiments in the literature. I myself have gone through the literature and tried to compare different studies, but the flow rate is often not reported. I think that intensive variables, such as emission factor (ug/g of oil used) or emission flux (ug/hour) would be more useful for comparison than air concentrations.

Thank you for your comment. Unlike vehicular emissions, there was no common sense about the emission rate or emission factor of cooking emissions (Atamaleki et al., 2021). Some studies indeed utilized mass concentration ($\mu g\ m^{-3}$) to demonstrate cooking emissions (Huang et al., 2020). We agree that the mass concentration varies between experiments and the flow rate of cooking fumes are unknown. We convert the mass concentration into emission rates ($\mu g\ min^{-1}$) for a better description of cooking emissions. Following is the revised sentence in the manuscript.

Emission rate (ER, $\mu g\ min^{-1}$) was calculated by the following equation, where $c$ is the blank subtracted mass concentration ($\mu g\ m^{-3}$) of the chemical quantified, and $Q$ is the mass flow of cooking exhaust emissions ($15\ m^3\ min^{-1}$).

$$ER = c \times Q \qquad\qquad (1)$$

We did the data treatment again and the results of ER, OFP, and SOA are all presented in the mass unit of $\mu g\ min^{-1}$.

Besides, we want to point out that though the cooking simulation experiments were conducted simultaneously, we sampled Tenax TA tubes without dilution, while (Zhang et al., 2021) sampled from diluted cooking fumes (dilution ratio: 8). The on- and off-line experiments were conducted

*separately*. Besides, the sampling time in this work is 15 ~ 30 min, not an hour.

In a related point, I am wondering how the quartz filter in front of the Tenax TA tubes affect the measurements, especially for the I/SVOCs. There are well known positive and negative artifacts for quartz filters, especially at high particle loadings. Some of the gaseous SVOCs can be lost to sorption onto filters (or organic material on filters), and some particle phase SVOCs could evaporate off the filter. What is the typical particle loading on these filters, and what is the potential for these artifacts to affect the SVOC measurements. This may be especially important for SOA estimation, if SVOCs contribute significantly to SOA.

Thank you for your comment. The emission patterns of particulate matters has been discussed in a another paper (Gong, Y., **Song, K.**, Guo, S., Lv, D., Zhang, Y., Wan, Z., Zhu, W., Wang, H., Yu, Y.,

Tan, R., Shen, R., Lu, S., Li, S., and Chen, Y.: Technical note: Identification and quantification of gaseous and particulate organic compounds from cooking fumes by comprehensive two-dimensional gas chromatography-mass spectrometry, Atmos. Chem. Phys. Discuss. [preprint], https://doi.org/10.5194/acp-2022-326, in review, 2022.). The total mass of particulate organics (29

mg m$^{-3}$) was much larger than the total mass of gaseous organics (2.4 mg m$^{-3}$). The particles mainly contains long-chain alkanoic and alkenoic acids (linoleic acid, oleic acid, and C16-mono-acid)

covering 82.5% of the total mass. The overlap of gas- and particle-phase organics are indeed small.

We added the uncertainty analysis in the manuscript as follows.

Cooking fumes were sampled directly without dilution. After collecting particles on quartz filters, gas-phase organics were sampled by pre-conditioned Tenax TA tubes (Gerstel 6 mm 97 OD,   4.5

mm   ID   glass   tube   filled   with   ~290   mg   Tenax   TA)   with   a   flow   of   0.5   L   min$^{-1}$.   The removal of particles on the quartz filter in front of the Tenax TA tubes affects the S/IVOC

measurements, causing positive and negative artifacts. Some of the gaseous SVOCs could be lost to sorption onto filters, and some particle-phase SVOCs could evaporate off the filter. The emission pattern of the particulate organics diverged from gas-phase organics, and a small overlap of species is identified. Aromatics, aldehydes, and short-chain acids mainly occurred in the gas-phase. For instance, the detection of short-chain olefinic aldehydes in the gas-phase was 40 times that of the particle-phase aldehydes. The artifacts of particulates on gas-phase aromatics and oxygenated compounds could be less than 5%. A typical system blank chromatogram is displayed in Figure S1. A

daily blank sampling of the air in the kitchen ventilator was conducted before cooking and was subtracted in the quantification procedure. All samples were frozen at -20°C before analyzing.

I am curious about the oil composition itself. Seems like it might be fairly straightforward to directly analyze the oil used, especially when answering the question about the differences in saturated and unsaturated fatty acid abundance. The type of oil (corn vs soybean vs other types) might not be as informative as the actual oil composition. Just a suggestion that would help add depth to the discussion, but I understand this will entail more experiments, so I will leave this up to the authors to decide whether this may be useful.

Thank you for your comment. We entailed a supplemental TD-GC×GC-MS experiment and added the information on oil molecular compositions in the manuscript.

Quartz filters added with about 1 mL of edible oils were also thermally desorbed and analyzed by TD-GC×GC-qMS. The total responses of blobs are normalized to 1 and the results were given by percent response (%).

…

Aromatic contributed 23.6%, 20.1%, 50.5%, and 19.8% of the total ERs of fried chicken fumes cooked with corn, peanut, soybean, and sunflower, oils, respectively. Fried chicken fumes cooked with soybean oil were especially abundant in toluene (rank 1st). In the TD-GC×GC-MS analysis of soybean oil (Figure S10), unsaturated fatty acids (linoleic acid) contributed 31.5% of the total percent response (50.5% aromatics), compared to 10.1% of the total response in corn oil (15.5%

aromatics). As a result, the aromatic concentrations and compositions of the fried chicken fumes diverged according to the content of unsaturated fatty acids in the oil (Chow, 2007; Zhang et al.,

2019).

…

Although pollutants were dominated by aromatics, alkanes, and oxygenated compounds with volatility bins of B9 to B12 (VOC-IVOC range, saturated vapor concentration $> 10^6$ μg m$^{-3}$) and polarity bins of P1 to P5 (low to medium polarity), significant diversities of volatility-polarity distributions were observed (Figure S9). The chemical compositions in each volatility bin were also distinct (Figure S11). IVOCs accounted for as much as 22.8% and 23.7% of the total ERs when peanut and sunflower oils were utilized for frying (Kostik et al., 2013; Ryan et al., 2008). The peanut oil was much more abundant in oleic acid (41.5%), while the proportion of linoleic acid in sunflower is 36.6% (Figure S10). The proportion of unsaturated acids in peanut and sunflower oils is higher than that of other oils.

[Figure]

Figure S10. Top 10 species in four edible oils (corn, peanut, soybean, and sunflower). Organics are normalized to 1 and the y-axis is the percent response (%).

I am also wondering how to interpret the main observations in the two different contexts: detailed composition and volatility/polarity distributions. The latter is a reductive approach to interpret complex organic composition, so it is not surprising that there can be larger differences in the composition (e.g. functional groups) between different experiments while the bulk volatility/polarity distribution stays relatively constant. Given the extensive analytical work performed in this study, it may be useful to dig deeper into what the composition changes can tell us. For example, is changing the oil changing the carbon number of the compounds (thereby changing the volatility distribution)

whereas the cooking style only changes the functional group (and perhaps replacing one functional group with another does not really impact volatility/polarity)?

Thank you for your comment. Volatility bins are commonly utilized in one-dimensional GC-MS. We want to stress that even though the volatility-polarity distribution is similar, the chemical composition could be largely different. We add figures elucidating the chemical compositions in each volatility bin in Figure S5 and Figure 12. We also revised the manuscript as follows.

Although the profiles of compositions diverged from dish to dish, their volatility-polarity patterns remained similar. The volatility-polarity distributions of the gaseous emissions are displayed in Figure 3. VOCs (B11 and before, saturated vapor concentration $> 10^6$ μg m$^{-3}$) with low polarity (P1 – P4) dominated the emissions of gas-phase contaminants. Chemicals in the VOC range accounted for 88.7%, 95.6%, 85.2%, and 81.4% of the total emission rates of fried chicken, Kung

Pao chicken, pan-fried tofu, and stir-fried cabbage emissions, while S/IVOCs accounted for 11.3%,

4.4%, 14.8%, and 18.2%, respectively. However, considering the chemical compositions in each volatility bin, the emission patterns are quite distinct (Figure S5). Oxygenated compounds were widely detected before B13 (VOC-IVOC range) in emissions of fried chicken and pan-fried tofu, while aromatics were extensively detected in the B8 range of Kung Pao chicken fumes. Alkanes and alkenes in the B10 range dominated the emissions of stir-fried cabbage. From the discussion above, the volatility distribution of cooking emissions obtained from the one-dimensional GC-MS analysis faces large uncertainty in SOA estimation if the polarity is not taken into account. Meanwhile, the volatility-polarity distribution should be equipped with detailed chemical parameters in each bin to precisely estimate SOA.

[Figure]

Figure S5. Chemical composition-volatility distributions of four dish emissions.

Although pollutants were dominated by aromatics, alkanes, and oxygenated compounds with volatility bins of B9 to B12 (VOC-IVOC range, saturated vapor concentration $> 10^6$ μg m$^{-3}$) and polarity bins of P1 to P5 (low to medium polarity), significant diversities of volatility-polarity distributions were observed (Figure S9). The chemical compositions in each volatility bin were also distinct (Figure S11). IVOCs accounted for as much as 22.8% and 23.7% of the total ERs when peanut and sunflower oils were utilized for frying (Kostik et al., 2013; Ryan et al., 2008). The peanut oil was much more abundant in oleic acid (41.5%), while the proportion of linoleic acid in sunflower is 36.6% (Figure S10). The proportion of unsaturated acids in peanut and sunflower oils is higher than that of other oils.

[Figure]

Figure S11. Chemical composition-volatility distributions of fried chicken emission cooked with four edible oils.

Similar to the previous comment, the authors made a claim in the concluding section:

*The PLS-DA and MPCA analysis indicated the importance of edible oils on cooking emissions. If*

*cooking-related pollution control strategies are made, the suggestion of deduction of oils that contain*

*more unsaturated fatty acids (such as soybean oil) could be taken into consideration.*

It seems to me that the conclusions from the PLS-DA and MPCA analysis concern the relative distributions, rather than absolute emissions. In other words, the analysis only tells you that the oil determine the variation in chemical composition, but not necessarily the amount of emissions. I do not disagree with the claim made in the manuscript; the evidence provided just does not support this claim.

Thank you for your comment. We have deleted the statement in the concluding section.

There also needs to be some discussion about the limitations of GC methods to comprehensively measure all compounds. Acids can decompose during thermal desorption, if no derivatization was performed. Highly polar compounds do not elute from the GC column. This may lead to biases in estimating polarity distributions.

Thank you for your comment. We add some uncertainty discussions in the *Atmospheric Implications* parts as follows.

We still need to stress that although GC×GC is utilized, UCMs still occur sharing a proportion of 5% of the total response in this work. Acids and aldehydes tail in the second column and cause uncertainties in the quantification procedure. Meanwhile, TD-GC×GC-MS does not comprehensively measure all compounds. Acids can decompose during thermal desorption if no derivatization was performed. Meanwhile, the decomposition of SVOC compounds could produce small molecules in the VOC or IVOC range. The TD process could introduce underestimation for SVOC compounds while causing overestimations of VOC and IVOC species. Highly polar compounds do not elute from the GC column. This may lead to biases in estimating volatility and polarity distributions. Comparisons between GC×GC and chemical ionization mass spectrometers (CIMS) should be further implemented to give a full glimpse of cooking organic compounds.

This work appears to be related to Zhang et al. ES&T 2021. How do the estimated SOA trends compare to AMS measurements? If the authors are able to reconcile SOA formation from AMS with bottom-up estimates from this work, it would allow us to assess how much we understand SOA formation in this system.

Thank you for your comment. This work is indeed related to Zhang et al. ES&T 2021 sharing the same lab, cooking material, cooking procedures, and edible oils. However, we sampled Tenax TA tubes without dilution, while (Zhang et al., 2021) sampled from diluted cooking fumes (dilution factor of 8). Besides, the on- and off-line experiments were conducted *separately*. The comparison between bottom-up and top-down methods is currently not available. However, in our recent publication (Yu et al., 2022), S/IVOCs are quantified by online VOCUS-PTR-ToF and the data were compared to AMS apportionment. VOCs could only explain 5% - 32% of the SOA, while this percentage rises to 19% - 55% when considering S/IVOC oxidation.

As mentioned earlier, I often find it difficult to understand what is being conveyed. The language in this manuscript is often confusing and awkward. There are also many instances of informal language that, in my view, is not consistent with scientific writing (e.g. "… is a tough job", "…better figure out…"). Furthermore, the number of significant figures in reported values is incongruent with the levels of uncertainty. While I will try to point out these instances of awkward language and inconsistent significant figures as much as I can in my detailed comments, there are far more than I

can point out individually, and much work is needed to resolve these issues.

Thank you for your comment. We go through the text carefully and have asked a native speaker to improve our writing.

Detailed comments:

Line 21: VOCs (not just S/IVOCs) are analyzed in this work too.

Thank you for your comment. The sentence is revised as follows.

**Volatile organic compounds (VOCs)**, intermediate volatility, and semi-volatile organic compounds (I/SVOCs) from cooking fumes were analyzed by a thermal desorption comprehensive two-dimensional gas chromatography coupled with quadrupole mass spectrometer (TD-GC×GC-qMS).

Line 66: "clarified" is an awkward word choice.

Thank you for your comment. The sentence is revised as follows.

Although chemical compositions, fingerprints, and influencing factors of cooking emissions have been **investigated** in some previous studies (Alves et al., 2021; Klein et al., 2016; Peng et al., 2017;

Vicente et al., 2021), there are still questions that remain uncertain.

Line 68: "constrain" is a verb, not a noun.

Thank you for your comment. The sentence is revised as follows.

The first **constraint** is that resolving complex mixtures of cooking emissions is rather tough.

Line 71: I am curious how speciating the UCM using GCxGC helped improved SOA estimation. In previous work, UCM is assigned SOA yields based on total signal and prescribed volatilities. So if that approach were used in this work, how different would that be from the more resolved estimates?

Thank you for your comment. Previous work has discussed the uncertainty of SOA estimation introduced by the bins-based method. The sentence in the manuscript is revised as follows.

It is of vital importance to identify chemical compositions of unresolved complex mixtures (UCM) to better understand their contributions to secondary organic aerosol (SOA). For instance, Huo et al investigated the S/IVOC emissions from incomplete combustion utilizing GC×GC-MS. They found that the previous bins-based method caused SOA underestimation with the ratio of 62.5 ±25.2% to 80.9 ±2.8% (Huo et al., 2021).

Line 73: "ones" is an awkward word choice.

Thank you for your comment. We have revised the manuscript as follows.

Particle-phase SVOC organics from cooking emissions are widely demonstrated yet few studies focus on gas-phase IVOC or SVOC **organics**.

Line 77: I am not sure that is quite true. The canonical studies from food cooking by Schauer et al. present very comprehensive profiles (Schauer et al., ES&T 1999).

Thank you for your comment. We have revised the manuscript as follows.

In other words, currently, there are few comprehensive source profiles of cooking emissions covering VOCs, IVOCs, and SVOCs (Schauer et al., 1999; Yu et al., 2022).

Line 112 and elsewhere: "comprehend" is not the correct word choice. Consider "understand" or "study".

Thank you for your comment. We have revised the manuscript as follows.

Meanwhile, four types of oil (i.e., soybean, corn, sunflower, and peanut oil ) were used for frying chicken to **illustrate** the influence of oil.

Line 115: it is slightly confusing to say that the emissions are mixed with ambient air (which is essentially dilution) and then say measured without dilution.

Thank you for your comment. We have deleted the statement about mixing and revised the manuscript as follows.

Cooking fumes were sampled directly without dilution.

Line 117: what are the breakthrough volumes of the most volatile compounds on the Tenax tubes?

0.5L/min for 90 minutes is about 45L. Are there concerns about compound breakthrough?

Thank you for your comment.

We did a supplementary experiment to examine the breakthrough effect by introducing pure nitrogen gas to the desorption tube with pre-added standard chemicals (Figure SS1). No significant breakthrough was observed within 24 h (<3%). The sampling time in this work is 15 ~ 30 min (0.5 L

min$^{-1}$) which is much less than 24h. We revised the manuscript as follows.

A Tenax TA breakthrough experiment was conducted by introducing pure nitrogen gas (N$_2$) with a flow of 0.5 L min$^{-1}$ to the desorption tube with pre-added standard chemicals (Figure S2). No significant breakthrough was observed within 24 h (<3% of TIC). The sampling time in this work is

15 ~ 30 min (0.5 L min$^{-1}$) which is much less than 24h.

[Figure]

Figure S2. The chromatograms of standard chemicals after 6h(brown), 24h(blue), 48h (red), and

72h (blue) of flowing by pure nitrogen gas. The flow of nitrogen gas is set to be the same as the sampling flow (0.5 L min$^{-1}$). No significant breakthrough was observed within 24 h (<3%).

Line 130: how good is the assumption that the 1$^{st}$ dimension retention time is representative of volatility? Did the authors verify against calculated vapor pressures?

Thank you for your comment. Pure liquid vapor pressure ($p_L$, Pa) has been estimated by pixel-based approaches in our previous publication (Song et al., 2022), which validates our statement.

Line 132: what does "qualified" mean?

Thank you for your comment. We have deleted the word as follows.

326 chemicals were quantified (Table S3) while 201 contaminants were detected (Table S4) in cooking fumes covering a wide range of VOCs, IVOCs, and SVOCs, including 25 aromatics, 19

$n$-alkanes, 100 oxygenated compounds (containing 7 acids, 10 alcohols, 29 aldehydes, 24 esters, 5

ketones, and others), 3 PAHs, and 54 other chemicals.

Line 132-133: "kinds" is an awkward word choice.

Thank you for your comment. We have revised the manuscript as follows.

326 chemicals were quantified (Table S3) while 201 contaminants were detected (Table S4) in cooking fumes covering a wide range of VOCs, IVOCs, and SVOCs, including 25 aromatics, 19

$n$-alkanes, 100 oxygenated compounds (containing 7 acids, 10 alcohols, 29 aldehydes, 24 esters, 5

ketones, and others), 3 PAHs, and 54 other chemicals.

Line 167: the word "form" is repeated. Also, I think the authors mean "format"?

Thank you for your comment. NetCDF is the abbreviation of Network Common Data Form. We have revised the manuscript as follows.

Chromatograms were imported from the network common data form (netCDF).

Line 190-193: how do these numbers compare to other works?

Thank you for your comment. We add a comparison as follows.

The compositions of the gaseous emissions are exhibited in Figure S4. Aromatics contributed

59.1%, 23.6%, 8.1%, and 11.8% of the total mass concentration of Kung Pao chicken, fried chicken, pan-fried tofu, and stir-fried cabbage, while oxygenated compounds accounted for 17.1%, 53.7%,

76.9%, and 25.0% of the total concentration, respectively. The compositions of organic in this study diverged from proton transfer reaction mass spectrometer (PTR-MS) measurements (Klein et al.,

2016; Liu et al., 2018), in which aldehydes dominated the emission profiles (~ 60%). The proportion of aromatics was also different from online Vocus-PTR-ToF measurements in a recent study (Yu et al., 2022). However, the contribution of aromatics was close to a recent study conducted at Chinese restaurants using GC-MS analysis (Huang et al., 2020). The different instruments resulting in different VOC detection ranges could be the explanation for the different patterns. GC$\times$GC-MS is powerful in resolving complex mixtures with carbon numbers of more than 6. The structural chromatograms and detailed mass spectrum information provide a convincing result in chemical identification (An et al., 2021). In contrast, PTR-MS could detect much more short-chain alkenes and aldehydes with carbon numbers less than 4. However, the isomers of PTR-MS could not be distinguished. Alkanes and some long-chain compounds could not be detected by PTR-MS. For instance, the maximum carbon number of pollutants in Yu et al is 16 ($C_{16}H_{26}$) (Yu et al., 2022) while the maximum carbon number of pollutants detected in this work is 30 ($C_{30}H_{62}$). $C_2H_6O$, $C_4H_{8,}$

$C_4H_8O2$, and $C_5H_8$ were the top species measured by Vocus-PTR-ToF (Yu et al., 2022), which is out of range of our measurement.

Line 209: It is more common in this field to use saturation vapor pressure or saturation concentrations to denote volatility, and O/C for polarity. What are the equivalent c* and O/C for these bins?

Thank you for your comment. We add the saturation concentrations in brackets as follows.

The volatility-polarity distributions of the gaseous emissions are displayed in Figure 3. VOCs (B11

and before, saturated vapor concentration $> 10^6$ $\mu g$ $m^{-3}$) with low polarity (P1 – P4) dominated the emissions of gas-phase contaminants.

Line 268-273: this paragraph is confusing. It may be helpful to have a sentence suggesting that this paragraph will be discussing the oil effect, rather than opening with "As for OFP estimation…"

Thank you for your comment. We revised the manuscript as follows.

Chicken fried in soybean oil produced the highest OFP ($10134 \pm 5958$ μg min$^{-1}$) while chicken fried in corn oil resulted in the most SOA estimation ($426 \pm 270$ μg min$^{-1}$).

Line 277: typo in "short-chain"

Thank you for your comment. We revised the manuscript as follows.

Despite the importance of aldehydes revealed in previous studies (Klein et al., 2016; Liu et al., 2018), our results demonstrated that alkanes, pinenes, and **short**-chain acids are also key precursors in cooking SOA production (Huang et al., 2020).

Line 279: what are "key reactions"? Is this referring to in-oil reactions? I am not sure if this study is really elucidating these reactions. Almost all cooking emission studies do not measure oil composition directly, and are only inferring these reactions based on food science literature. It is unclear if these measurements help elucidate these reactions.

Thank you for your comment. We revised the subtitle as follows.

3.3 Elucidating the influencing factor and **inferring** in-oil reactions of cooking emissions

Line 289: typo in "variance"

Thank you for your comment. We revised the manuscript as follows.

The **variance** of cooking fumes could be largely explained by the different oil utilized.

Line 304-306: this is an interesting point. Did the emissions of aromatics increase with degree of unsaturation in oil?

Thank you for your comment. The emissions of aromatics decrease with the decreasing degree of unsaturation in oil. We revised the subtitle as follows.

In more detail, the oxidation of unsaturated fatty acids (such as linoleic acid) in oil leads to the production of alkadienals (such as (*E*, *E*)-2,4-decadienal) which form aromatics (butylbenzene) by losing $H_2O$ (Atamaleki et al., 2021; Zhang et al., 2019). This is consistent with the analysis of edible oils in this work. The emission pattern is in line with previous studies (Atamaleki et al., 2021). Corn oil contained a less amount of unsaturated fatty acids (Figure S10), and the emission of aromatics cooked with corn oil was the lowest among the 4 types of oils used. The emission pattern is in line with previous studies (Atamaleki et al., 2021).

Section 4: the conclusion section is more a recap of the results and discussion, and very thin on implications and limitations. I suggest a broader discussion of context, and posing future research questions.

Thank you for your comment. We revised section 4 as follows.

[revised manuscript text omitted]

Line 322-323: the authors can substantiate this claim with much more quantitative information. How much of the estimated SOA is from aldehydes versus other compounds based on the calculated SOA

formation potential (equation 2)?

Thank you for your comment. We revised the manuscript as follows.

The influence of cooking style and oil is taken into consideration in this work. S/IVOC species are key components as they contributed 10.2% - 32.0% of the total SOA estimation.

Aldehydes only accounted for 0.6% -10.1% of the total SOA estimation. We revised the manuscript as follows.

Previous works might underestimate the importance of cooking fumes to SOA formation because only a series of IVOC homologs were quantified (Liu et al., 2018). For instance, aldehydes only accounted for 0.6% -10.1% of the total SOA estimation. If only aldehydes are taken into consideration, SOA will be underestimated 9.9 ~139 times.

Supplemental Information:

Table S1: how were oil temperatures measured or estimated?

Thank you for your comment. The oil temperature was measured by a thermometer placed in the oil.

The thermometer was removed from the oil before placing the cooking materials. We revised Table

S1 as follows.

**Table S1.** Details of cooking procedures.

| Domestic cooking | Material | Oil temperature [#] |
|---|---|---|
| Fried chicken | 170 g chicken, 500 mL oil (corn, peanut, soybean, or sunflower oil), a few condiments | 145 ~ 150 ℃ |
| Kung Pao chicken | 150 g chicken, 50 g peanut, 40 mL corn oil, a few condiments | Not stable |

| | | |
|---|---|---|
| Pan-fried tofu | 500 g tofu, 200 mL corn oil, a few condiments | 100 ~ 110 ℃ |
| Stir-fried cabbage | 300 g chicken, 40 mL corn oil, a few condiments | 95 ~ 105 ℃ |

[#] The oil temperature was measured by a thermometer placed in the oil. The thermometer was removed from the oil before placing the cooking materials. The temperatures listed in Table S1 were initial cooking temperatures and were maintained the same for each dish.

**Reference:**

[revised manuscript text omitted]

---

## Author Comment (AC2)

We thank the reviewers for their careful review of the manuscript. The comments greatly improved our manuscript. We revised our manuscript according to the reviewers' comments and suggestions. Overall, we have changed the mass concentration ($\mu g\ m^{-3}$) to the emission rate ($\mu g\ min^{-1}$) to avoid the influence of cooking time and sampling time according to the comments of the referees. We add more details to the volatility distributions of cooking emissions. We also added more comparisons with different studies. Following are our responses to the comments.

**Response to referee #2:**

General comments:

Cooking emissions are an important source of primary and secondary organic aerosols in the urban environment. However, detailed speciation of non-methane organic gases (NMOGs) emitted from food cooking is lacking. In this study, Song et al. characterized the VOCs and S/IVOCs from cooking typical Chinese dishes using a TD-GC×GC-qMS. They found that the volatility-polarity distributions of gaseous organic species from four dishes were similar. S/IVOCs were predicted to contribute as high as 32% of the estimated SOA formation. The variations of chemical compositions of NMOGs were mainly caused by the cooking oils instead of cooking styles. This paper provides important information to the atmospheric chemistry and air quality community. However, the conclusions are inconsistent with a recent paper published by the same research group (Yu et al., 2022). For example, this study found that aromatics contributed around 59% of the NMOG emissions from kung pao chicken while only a small fraction was reported by Yu et al. 2022. More discussions and clarifications are needed to address the differences between these two studies. Also, the language should be edited and polished. I recommend this paper be published after addressing the following comments.

Thank you for your comments. We have asked a native speaker to go through our text. Following are our point-to-point responses to the comments.

Specific comments:

The mass concentrations of NMOGs were compared for different dishes. However, the mass concentrations highly depended on the cooking time and sampling time for each dish. Emission rates
(mg/min) or emission factors (mg/kg) are more appropriate for comparison of emissions from
cooking different dishes.

Thank you for your comment. Unlike vehicular emissions, there was no common sense about the
emission rate or emission factor of cooking emissions (Atamaleki et al., 2021). Some studies indeed
utilized mass concentration ($\mu g\ m^{-3}$) to demonstrate cooking emissions (Huang et al., 2020). We
agree that the mass concentration varies between experiments and the flow rate of cooking fumes are
unknown. We convert the mass concentration into emission rates ($\mu g\ min^{-1}$) for a better description
of cooking emissions. Following is the revised sentence in the manuscript.

Emission rate (ER, $\mu g\ min^{-1}$) was calculated by the following equation, where $c$ is the blank
subtracted mass concentration ($\mu g\ m^{-3}$) of the chemical quantified, and $Q$ is the mass flow of cooking
exhaust emissions (15 $m^3\ min^{-1}$).

$$ER = c \times Q \qquad\qquad (1)$$

We did the data treatment again and the results of ER, OFP, and SOA are all presented in the mass
unit of $\mu g\ min^{-1}$.

The chemical composition of NMOGs for cooking the same dish determined
using TD-GC×GC-qMS in this study is inconsistent with that determined using VOCUS-PTR-ToF
despite that VOCUS cannot measure alkanes (Yu et al., 2022). TD-GC×GC-qMS detected more
aromatics while VOCUS detected more aldehydes. Why is there such a big difference?

Thank you for your comment. This work is indeed related to Zhang et al. EST 2021 and Yu et al.,
ESTL 2022 sharing the same lab, cooking material, cooking procedures, and edible oils. However,
we sampled Tenax TA tubes without dilution, while (Zhang et al., 2021) sampled from diluted
cooking fumes (dilution factor of 8). Besides, the on- and off-line experiments were conducted
*separately*. The detection range of TD-GC×GC-qMS and Vocus-PTR-ToF also diverged, as the
short-chain alkenes, and acids are missing in this work, while long-chain S/IVOCs (<C16), alkanes
are not detected in Yu et al., We added more detail as follows.

The compositions of the gaseous emissions are exhibited in Figure S4. Aromatics contributed

59.1%, 23.6%, 8.1%, and 11.8% of the total mass concentration of Kung Pao chicken, fried chicken, pan-fried tofu, and stir-fried cabbage, while oxygenated compounds accounted for 17.1%, 53.7%, 76.9%, and 25.0% of the total concentration, respectively. The compositions of organic in this study diverged from proton transfer reaction mass spectrometer (PTR-MS) measurements (Klein et al., 2016; Liu et al., 2018), in which aldehydes dominated the emission profiles (>60%). The content of aromatics was also different from online Vocus-PTR-ToF measurements in a recent study (Yu et al., 2022). However, the contribution of aromatics was close to a recent study conducted at Chinese restaurants using GC-MS analysis (Huang et al., 2020). The different instruments resulting in different VOC detection ranges could be the explanation for the different patterns. GC×GC-MS is powerful in resolving complex mixtures with carbon numbers of more than 6. The structural chromatograms and detailed mass spectrum information provide a convincing result in chemical identification (An et al., 2021). In contrast, PTR-MS could detect much more short-chain alkenes and aldehydes with carbon numbers less than 4. However, the isomers of PTR-MS could not be distinguished. Alkanes and some long-chain compounds could not be detected by PTR-MS. For instance, the maximum carbon number of pollutants in Yu et al is 16 ($C_{16}H_{26}$) (Yu et al., 2022) while the maximum carbon number of pollutants detected in this work is 30 ($C_{30}H_{62}$). $C_2H_6O$, $C_4H_{8,}$ $C_4H_8O2$, and $C_5H_8$ were the top species measured by Vocus-PTR-ToF (Yu et al., 2022), which is out of range of our measurement.

The SOA formation potential was estimated by assuming a yield for the potential SOA precursors, which may introduce large uncertainties to the estimation. For example, acetic acid (Table S3) was regarded as an SOA precursor. However, no studies reported that the oxidation of acetic acid can produce SOA. The VOCs used for SOA estimations should have been identified as SOA precursors by previous studies. Also, the SOA estimations are insistent with the measurements by Yu et al. (2022). This study estimated that Kung Pao chicken would produce the highest SOA mass while Yu et al. (2022) measured that Kung Pao chicken formed the lowest SOA mass. The authors should discuss why the estimations are inconsistent with the measurements.

Thank you for your comment. We have double-checked Table S3 and remove those unconvincing yields. We also add references to Table S3.

We also compared the results of SOA estimation and measurement with Yu et al. Large uncertainties remain in SOA estimation. Yu et al measured gas-phase VOC, IVOC, and SVOC

precursors by Vocus-PTR-ToF and compared the results with SOA measured from the aerosol mass spectrometer (AMS). 19 ~ 55% of the SOA could be explained. Among them, the SOA estimation from precursors emitted from Kung Pao chicken is the largest even though the SOA mass is the lowest among the four dished (Yu et al., 2022). The SOA estimation in this work is also the largest regarding Kung Pao chicken emissions. Aromatics and alkenes in Kung Pao chicken fumes contributed 63.6% of the SOA estimation, and the top SOA contributor in Yu et al. were sesquiterpenes and aromatics, showing a consistent pattern between these two studies. It should be noticed that more than 45% of the SOA could not be explained (Yu et al., 2022) and more investigations should be carried on to further identify the emission and evolution of cooking fumes in the atmosphere.

The total emission rates, compositions, and volatility-polarity distributions of OFP and SOA

estimation by gaseous precursors are displayed in Figure 2, Figure S4, and Figure 3, respectively.

The total OFP and SOA estimation are consistent with the emission rate, as Kung Pao chicken emitted the most pollutants and produced the most ozone formation ($21125 \pm 19447$ μg min$^{-1}$) and

SOA formation ($584 \pm 482$ μg min$^{-1}$). Pan-fried tofu emitted a little bit less than fried chicken, yet produced more SOA estimation due to a large proportion of short-chain acids (hexanoic acid) (Alves and Pio, 2005; Forstner et al., 1997; Kamens et al., 1999). Short-chain acids are likely derived from scission reactions of allylic hydroperoxides originating from unsaturated fatty acids (Chow, 2007;

Goicoechea and Guillén, 2014). Although chemicals in the VOC range dominated ozone and SOA

formation, an increase in ozone formation contribution and a decrease in SOA formation contribution compared with the mass proportion of VOCs in ERs were observed. VOCs contributed 90.3% - 99.8%

of the ozone estimation, and 68.0% - 89.8% of the total SOA estimation, compared with 81.4% -

95.6% in ERs. S/IVOCs explained 10.2% - 32.0% of the SOA estimation. Aromatics (toluene) and alkenes (heptene) were dominant ozone formation precursors in meat-relating dishes (fried chicken,

Kung Pao chicken, and pan-fried tofu), while alcohols (butanol and linalool) were predominant for stir-fried cabbage (Atamaleki et al., 2021). Acids (hexanoic acid), aromatics (toluene), alkenes (pinenes), and alkanes were important SOA precursors. We also want to emphasize that there are large uncertainties in SOA estimation. Yu et al measured gas-phase VOC, IVOC, and SVOC precursors by Vocus-PTR-ToF and compared the results with SOA measured from the aerosol mass spectrometer (AMS). 19 ~ 55% of the SOA could be explained. Among them, the SOA estimation from precursors emitted from Kung Pao chicken is the largest even though the SOA mass is the lowest among the four dished (Yu et al., 2022). The SOA estimation in this work is also the largest regarding Kung Pao chicken emissions. Aromatics and alkenes in Kung Pao chicken fumes contributed 63.6% of the SOA estimation, and the top SOA contributor in Yu et al. were sesquiterpenes and aromatics, showing a consistent pattern between these two studies. It should be noticed that more than 45% of the SOA could not be explained (Yu et al., 2022) and more investigations should be carried on to further identify the emission and evolution of cooking fumes in the atmosphere.

Lines 27-28: The authors stated that "Dishes cooked by stir-frying or deep-frying cooking styles emit much more pollutants than relatively mild cooking methods". However, this is not supported by the measurement. Figure S3 shows that stir-frying cabbage emitted the lowest amount of gaseous species. Which dish was cooked in a mild style? Is it pan-fried tofu?

Thank you for your comment. We have revised the manuscript as follows.

Kung Pao chicken emitted more pollutants than other dishes due to its rather intense cooking method.

Lines 116-117: It is helpful to provide the sampling procures of the Tenax tubes. Is there a breakthrough?

Thank you for your comment.

We did a supplementary experiment to examine the breakthrough effect by introducing pure nitrogen gas to the desorption tube with pre-added standard chemicals (Figure SS1). No significant breakthrough was observed within 24 h (<3%). The sampling time in this work is 15 ~ 30 min (0.5 L

min$^{-1}$) which is much less than 24h. We revised the manuscript as follows.

A Tenax TA breakthrough experiment was conducted by introducing pure nitrogen gas ($N_2$) with a flow of 0.5 L min$^{-1}$ to the desorption tube with pre-added standard chemicals (Figure S2). No significant breakthrough was observed within 24 h (<3% of TIC). The sampling time in this work is

15 ~ 30 min (0.5 L min$^{-1}$) which is much less than 24h.

[Figure]

Figure S2. The chromatograms of standard chemicals after 6h(brown),24h(blue),48h (red), and

72h (blue) of flowing by pure nitrogen gas. The flow of nitrogen gas is set to be the same as the sampling flow (0.5 L min$^{-1}$). No significant breakthrough was observed within 24 h (<3%).

Line 183: Figure S3 displays one of the main results. It should go to the main paper. The unit of the y axis is missing.

Thank you for your comments. We moved the figure to the main paper and added the unit of the

*y*-axis.

[Figure]

Figure 2. Emission rate (ER), ozone formation potential (OFP), and secondary organic aerosol (SOA)

estimation from emissions of fried chicken, Kung Pao chicken, pan-fried tofu, and stir-fried cabbage.

The unit of the *y*-axis is μg min$^{-1}$.

Lines 217-219: Is there any evidence that these small acids can produce SOA?

Thank you for your comments. We have added references to this statement. Besides, the mechanism of propanoic acid (C3-mono acid) oxidation has been added to the MCM model (http://chmlin9.leeds.ac.uk/MCMv3.3.1/roots.htt).

Pan-fried tofu emitted a little bit less than fried chicken, yet produced more SOA estimation due to a large proportion of short-chain acids (hexanoic acid) (Alves and Pio, 2005; Forstner et al., 1997;

Kamens et al., 1999).

Line 235: I would suggest moving Figure S7 to the main paper.

Thank you for your comments. We moved the figure to the main paper and added the unit of the y-axis.

[Figure]

Figure 4. Emission rate (ER), ozone formation potential (OFP), and secondary organic aerosol (SOA)

estimation from emissions of fried chicken cooked with corn, peanut, soybean, and sunflower oils.

The unit of the y-axis is μg min$^{-1}$.

Lines 319-320: I would suggest removing this statement as Yu et al. (2022) already characterized the

S/IVOCs from food cooking.

Thank you for your comments. We have revised the manuscript as follows.

In this work, gaseous VOCs, IVOCs, and SVOCs from cooking fumes are quantified in detail.

Technical corrections:

Line 66: Please consider changing "clarified" to "investigated" or "studied".

Thank you for your comment. The sentence is revised as follows.

Although chemical compositions, fingerprints, and influencing factors of cooking emissions have been **investigated** in some previous studies (Alves et al., 2021; Klein et al., 2016; Peng et al., 2017;

Vicente et al., 2021), there are still questions that remain uncertain.

Table S3: Please list the reference for estimating the SOA yield of each compound.

Thank you for your comment. Table S3 is revised as follows.

Table S3. Chemials quantified, with chemical classes, $R^2$, MIR, kOH, yield, surrogates and references. The SOA yields of precursors were from literature (Algrim and Ziemann, 2016, 2019; Chan et al., 2009, 2010; Harvey and Petrucci, 2015; Li et al., 2016; Liu et al., 2018; Loza et al., 2014; Matsunaga et al., 2009; McDonald et al., 2018; Shah et al., 2020; Tkacik et al., 2012; Wu et al., 2017) or surrogates from *n*-alkanes in the same volatility bins (Zhao et al., 2014, 2017).

| compound | class detail | class | $R^2$ | MIR | OFP surrogate | kOH | kOH_reference | yield | yield_surrogate | Yield_reference |
|---|---|---|---|---|---|---|---|---|---|---|
| C6 | alkanes | alkanes | 0.98 | 1.24 | | 5.20 | Atkinson and Arey,2003 | 0.00 | | Wu et al., 2017 |
| C7 | alkanes | alkanes | 0.98 | 1.07 | | 6.76 | Atkinson and Arey,2003 | 0.05 | | Wu et al., 2017 |
| b-alkanes-C10 | b-alkanes | alkanes | 0.92 | 0.68 | C10 | | | 0.22 | C10 | |
| b-alkanes-C11 | b-alkanes | alkanes | 0.90 | 0.61 | C11 | | | 0.33 | C11 | |
| b-alkanes-C12 | b-alkanes | alkanes | 0.99 | 0.55 | C12 | | | 0.02 | C12 | |
| b-alkanes-C13 | b-alkanes | alkanes | 0.94 | 0.53 | C13 | | | 0.03 | C13 | |
| b-alkanes-C14 | b-alkanes | alkanes | 0.93 | 0.51 | C14 | | | 0.05 | C14 | |
| b-alkanes-C15 | b-alkanes | alkanes | 0.98 | 0.50 | C15 | | | 0.08 | C15 | |
| b-alkanes-C16 | b-alkanes | alkanes | 0.95 | 0.45 | C16 | | | 0.12 | C16 | |
| b-alkanes-C17 | b-alkanes | alkanes | 0.92 | 0.42 | C17 | | | 0.20 | C17 | |
| b-alkanes-C18 | b-alkanes | alkanes | 0.96 | 0.40 | C18 | | | 0.30 | C18 | |
| b-alkanes-C19 | b-alkanes | alkanes | 0.89 | 0.38 | C19 | | | 0.42 | C19 | |
| b-alkanes-C20 | b-alkanes | alkanes | 0.95 | 0.36 | C20 | | | 0.56 | C20 | |
| Heptane, 2-methyl- | b-alkanes | alkanes | 0.76 | 1.07 | | 8.28 | AopWin | 0.06 | C8 | |
| b-alkanes-C8 | b-alkanes | alkanes | 0.76 | 0.90 | C8 | | | 0.06 | C8 | |
| b-alkanes-C9 | b-alkanes | alkanes | 0.92 | 0.78 | C9 | | | 0.14 | C9 | |
| Cyclohexane, propyl- | cyclo-alkanes | alkanes | 0.92 | 1.29 | | 13.40 | AopWin | 0.14 | C9 | |
| Cyclopentane, butyl- | cyclo-alkanes | alkanes | 0.92 | 1.29 | Cyclohexane, propyl- | | | 0.14 | C9 | |
| Bicyclo[5.3.0]decane | cyclo-alkanes | alkanes | 0.92 | 1.29 | Cyclohexane, propyl- | | | 0.22 | C10 | |

| | | | | | | | | | | |
|---|---|---|---|---|---|---|---|---|---|---|
| es | | | | | | | | | | |
| **Cyclohexene, 3-methyl-6-(1-methyl ethyl)-, trans-** | cyclo-alkanes | alkanes | 0.92 | 1.29 | Cyclohexane, propyl- | | | 0.22 | C10 | |
| **Cyclohexene, 4-propyl-** | cyclo-alkanes | alkanes | 0.92 | 1.29 | Cyclohexane, propyl- | | | 0.14 | C9 | |
| **Cyclopentene,3-hexyl-** | cyclo-alkanes | alkanes | 0.90 | 1.29 | Cyclohexane, propyl- | | | 0.33 | C11 | |
| **alkenes-C12** | n-alkanes | alkanes | 0.99 | 1.48 | alkenes-C13 | | | 0.47 | | Matsunaga, Aiko,2009 |
| **3-Dodecene, (E)-** | n-alkanes | alkanes | 0.99 | | | | | 0.47 | alkenes-C12 | |
| **alkenes-C13** | n-alkanes | alkanes | 0.94 | 1.48 | | 40.07 | AopWin | 0.46 | | Matsunaga, Aiko,2009 |
| **alkenes-C14** | n-alkanes | alkanes | 0.93 | 1.34 | | 41.48 | AopWin | 0.50 | | Matsunaga, Aiko,2009 |
| **alkenes-C15** | n-alkanes | alkanes | 0.98 | 1.25 | | 42.90 | AopWin | 0.53 | | Matsunaga, Aiko,2009 |
| **alkenes-C16** | n-alkanes | alkanes | 0.95 | 1.25 | alkenes-C15 | | | 0.64 | | Matsunaga, Aiko,2009 |
| **alkenes-C17** | n-alkanes | alkanes | 0.92 | 1.25 | alkenes-C15 | | | 0.49 | | Matsunaga, Aiko,2009 |
| **alkenes-C18** | n-alkanes | alkanes | 0.96 | 1.25 | alkenes-C15 | | | 0.49 | alkenes-C17 | |
| **C7** | n-alkanes | alkanes | 0.98 | 1.24 | | 5.20 | Atkinson and Arey,2003 | 0.00 | | Wu et al., 2017 |
| **C8** | n-alkanes | alkanes | 0.98 | 0.90 | | 8.11 | Atkinson and Arey,2003 | 0.06 | | Wu et al., 2017 |
| **C9** | n-alkanes | alkanes | 1.00 | 0.78 | | 9.70 | Atkinson and Arey,2003 | 0.14 | | Wu et al., 2017 |
| **C10** | n-alkanes | alkanes | 0.99 | 0.68 | | 11.00 | Atkinson and Arey,2003 | 0.22 | | Wu et al., 2017 |
| **C11** | n-alkanes | alkanes | 0.97 | 0.61 | | 12.30 | Atkinson and Arey,2003 | 0.33 | | Wu et al., 2017 |
| **C12** | n-alkanes | alkanes | 0.99 | 0.55 | | 13.20 | Atkinson and Arey,2003 | 0.02 | | Chan et al., 2009 |
| **C13** | n-alkanes | alkanes | 0.98 | 0.53 | | 15.10 | Atkinson and Arey,2003 | 0.03 | | Chan et al., 2009 |
| **C14** | n-alkanes | alkanes | 0.99 | 0.51 | | 17.90 | Atkinson and Arey,2003 | 0.05 | | Chan et al., 2009 |
| **C15** | n-alkanes | alkanes | 0.99 | 0.50 | | 20.70 | Atkinson and Arey,2003 | 0.08 | | Chan et al., 2009 |
| **C16** | n-alkanes | alkanes | 0.99 | 0.45 | | 23.20 | Atkinson and Arey,2003 | 0.12 | | Chan et al., 2009 |
| **C17** | n-alkanes | alkanes | 0.99 | 0.42 | | 28.50 | A. W. H. Chan et al,2009 | 0.20 | | Chan et al., 2009 |
| **C18** | n-alkanes | alkanes | 0.99 | 0.40 | | 35.10 | A. W. H. Chan et al,2009 | 0.30 | | Chan et al., 2009 |
| **C19** | n-alkanes | alkanes | 0.99 | 0.38 | | 43.20 | A. W. H. Chan et al,2009 | 0.42 | | Chan et al., 2009 |

| | | | | | | | | | | |
|---|---|---|---|---|---|---|---|---|---|---|
| C20 | n-alkanes | alkanes | 0.99 | 0.36 | | 53.10 | A. W. H. Chan et al,2009 | 0.56 | | Chan et al., 2009 |
| C21 | n-alkanes | alkanes | 1.00 | 0.34 | | 26.65 | AopWin v1.92 | 0.77 | | Gentner, 2012 |
| C22 | n-alkanes | alkanes | 1.00 | 0.33 | | 28.07 | AopWin v1.92 | 0.96 | | Gentner, 2012 |
| C23 | n-alkanes | alkanes | 1.00 | | | 29.48 | AopWin v1.92 | 1.08 | | Gentner, 2012 |
| C24 | n-alkanes | alkanes | 1.00 | | | 30.89 | AopWin v1.92 | 1.14 | | Gentner, 2012 |
| C26 | n-alkanes | alkanes | 1.00 | | | 33.72 | AopWin v1.92 | 1.14 | C24 | |
| C27 | n-alkanes | alkanes | 0.99 | | | 35.13 | AopWin v1.92 | 1.14 | C24 | |
| C30 | n-alkanes | alkanes | 1.00 | | | 39.37 | AopWin v1.92 | 1.14 | C24 | |
| alk-di-enes-C12 | alkenes | alkenes | 0.99 | | | | | 0.41 | alpha-Pinene | |
| 1-Heptene | alkenes | alkenes | 0.95 | 4.43 | | 40.00 | Atkinson and Arey,2003 | 0.02 | | Wu et al., 2017 |
| 1-Octene | alkenes | alkenes | 0.76 | 3.25 | | 33.00 | AopWin | 0.05 | | Matsunaga, Aiko,2009 |
| 2-Octene, (E)- | alkenes | alkenes | 0.76 | 6.00 | | 61.83 | AopWin | 0.05 | 1-Octene | |
| 3-Nonene | alkenes | alkenes | 0.92 | 6.00 | 2-Octene, (E)- | | | 0.15 | 1-Nonene | |
| 1-Nonene | alkenes | alkenes | 0.92 | 2.60 | | 34.42 | AopWin | 0.15 | | |
| 2-Nonene | alkenes | alkenes | 0.92 | 6.00 | 2-Octene, (E)- | | | 0.15 | 1-Nonene | |
| 1,3-Nonadiene, (E)- | alkenes | alkenes | 0.92 | 2.17 | 1-Decene | | | 0.15 | 1-Nonene | |
| 1-Decene | alkenes | alkenes | 0.92 | 2.17 | | 35.83 | AopWin | 0.32 | | Matsunaga, Aiko,2009 |
| trans-3-Decene | alkenes | alkenes | 0.92 | | | | | 0.32 | 1-Decene | |
| Dicyclopentadiene | alkenes | alkenes | 0.92 | | | | | 0.34 | 1-Undecene | Matsunaga, Aiko,2009 |
| 1,10-Undecadiene | alkenes | alkenes | 0.90 | 2.17 | 1-Decene | | | 0.34 | 1-Undecene | |
| 4-Undecene, (E)- | alkenes | alkenes | 0.90 | 6.00 | 2-Octene, (E)- | | | 0.34 | 1-Undecene | |
| trans,trans-2,9-Undecadiene | alkenes | alkenes | 0.90 | | | | | 0.34 | 1-Undecene | |
| 2-Undecene, (E)- | alkenes | alkenes | 0.90 | 6.00 | 2-Octene, (E)- | | | 0.34 | 1-Undecene | |
| 2-Undecene, (Z)- | alkenes | alkenes | 0.90 | 6.00 | 2-Octene, (E)- | | | 0.34 | 1-Undecene | |
| (E,E)-1,3,5-Undecatriene | alkenes | alkenes | 0.99 | 2.17 | 1-Decene | | | 0.34 | 1-Undecene | |
| 1,8,11-Heptadecatrien | alkenes | alkenes | 0.92 | 2.17 | 1-Decene | | | 0.49 | alkenes-C17 | |

| | | | | | | | | | | |
|---|---|---|---|---|---|---|---|---|---|---|
| e, (Z,Z)- | | | | | | | | | | |
| **alkenes-C17-UCM** | alkenes | alkenes | 0.92 | 2.17 | 1-Decene | | | 0.49 | alkenes-C17 | |
| **di-isoprenens** | di-isoprenes | alkenes | 0.92 | 4.51 | alpha-Pinene | | | 0.41 | alpha-Pinene | |
| **4,7-Methano-1H-indene, octahydro-,** | di-isoprenes | alkenes | 0.90 | | | | | 0.41 | alpha-Pinene | |
| **Bicyclo[3.1.0]hex-2-ene, 2-methyl-5-(1-methyl ethyl)-** | di-isoprenes | alkenes | 0.92 | 4.51 | alpha-Pinene | | | 0.41 | alpha-Pinene | |
| **Bicyclo[3.1.0]hex-2-ene, 4-methyl-1-(1-methyl ethyl)-** | di-isoprenes | alkenes | 0.92 | 4.51 | alpha-Pinene | | | 0.41 | alpha-Pinene | |
| **alpha-Pinene** | di-isoprenes | alkenes | 0.92 | 4.51 | | 52.30 | Atkinson and Arey,2003 | 0.41 | | Lee et al., 2006 |
| **beta-Pinene** | di-isoprenes | alkenes | 0.92 | 3.52 | | | | 0.22 | C10 | |
| **beta-Myrcene** | di-isoprenes | alkenes | 0.92 | 4.51 | alpha-Pinene | 215.00 | Atkinson and Arey,2003 | 0.11 | | Lee et al., 2006 |
| **D-Limonene** | di-isoprenes | alkenes | 0.92 | 4.55 | | 164.00 | Atkinson | 0.41 | alpha-Pinene | |
| **di-isoprenes** | di-isoprenes | alkenes | 0.92 | 4.51 | alpha-Pinene | | | 0.22 | C10 | |
| **trans-beta-Ocimene** | di-isoprenes | alkenes | 0.92 | | | 252.00 | Atkinson and Arey,2003 | 0.41 | alpha-Pinene | |
| **1,3,6-Octatriene, 3,7-dimethyl-, (Z)-** | di-isoprenes | alkenes | 0.92 | 4.51 | alpha-Pinene | 252.00 | Atkinson and Arey,2003 | 0.41 | alpha-Pinene | |
| **Cyclohexene,** | di-isoprenes | alkenes | 0.90 | 6.36 | | 225.0 | Atkinson and Arey,2003 | 0.20 | | Lee et al., 2006 |

| Compound | | | | | | | | | | |
|---|---|---|---|---|---|---|---|---|---|---|
| **1-methyl-4-(1-methyl ethylidene)-** | s | | | | | 0 | | | | |
| **Copaene** | tri-isoprenes | alkenes | 0.93 | | | 90.00 | Atkinson and Arey,2003 | 0.41 | alpha-Pinene | |
| **Longifolene** | tri-isoprenes | alkenes | 0.93 | | | 47.00 | Atkinson and Arey,2003 | 0.41 | alpha-Pinene | |
| **alpha-Patchoulene** | tri-isoprenes | alkenes | 0.95 | | | | | 0.41 | alpha-Pinene | |
| **tri-isoprenes** | tri-isoprenes | alkenes | 0.95 | | | | | 0.41 | alpha-Pinene | |
| **3-Nonen-1-yne, (E)-** | alkynes | alkynes | 0.92 | | | | | 0.15 | 1-Nonene | |
| **alkynes-C12** | n-alkynes | alkynes | 0.99 | | | | | 0.47 | alkenes-C12 | |
| **alkynes-C13** | n-alkynes | alkynes | 0.94 | | | | | 0.46 | alkenes-C13 | |
| **alkynes-C14** | n-alkynes | alkynes | 0.93 | | | | | 0.50 | alkenes-C14 | |
| **alkynes-C15** | n-alkynes | alkynes | 0.98 | | | | | 0.53 | alkenes-C15 | |
| **alkynes-C16** | n-alkynes | alkynes | 0.95 | | | | | 0.64 | alkenes-C16 | |
| **alkynes-C17** | n-alkynes | alkynes | 0.92 | | | | | 0.49 | alkenes-C17 | |
| **alkynes-C18** | n-alkynes | alkynes | 0.96 | | | | | 0.49 | alkenes-C17 | |
| **Toluene** | aromatics | aromatics | 0.94 | 4.00 | | 5.63 | Atkinson and Arey,2003 | 0.10 | | Chan et al., 2009 |
| **Ethylbenzene** | aromatics | aromatics | 0.89 | 3.04 | | 7.00 | Atkinson and Arey,2003 | 0.10 | | Chan et al., 2009 |
| **p-Xylene** | aromatics | aromatics | 0.87 | 5.84 | | 14.30 | Atkinson and Arey,2003 | 0.06 | | Chan et al., 2009 |
| **Styrene** | aromatics | aromatics | 0.71 | 1.73 | | 58.00 | Atkinson and Arey,2003 | 0.22 | | Fang et al., 2017 |
| **o-xylene** | aromatics | aromatics | 0.71 | 5.84 | p-Xylene | | | 0.06 | *p*-Xylene | |
| **Benzene, (1-methylethyl)-** | aromatics | aromatics | 0.98 | 2.52 | | 6.30 | Atkinson and Arey,2003 | 0.03 | | Li et al., 2016 |
| **Benzene, 1-ethyl-4-methyl-** | aromatics | aromatics | 0.63 | 4.44 | | 11.80 | Atkinson and Arey,2003 | 0.10 | | Chan et al., 2009 |
| **Benzene, 1,2,3-trimethyl-** | aromatics | aromatics | 0.63 | 11.97 | | 32.70 | Atkinson and Arey,2003 | 0.08 | | Li et al., 2016 |

| | | | | | | | | | |
|---|---|---|---|---|---|---|---|---|---|
| Benzene, 1-ethyl-2-methyl- | aromatics | aromatics | 0.63 | 7.39 | Benzene, 1-ethyl-3-methyl- | | | 0.08 | Benzene, 1-ethyl-2-methyl- |
| Benzene, 1,2,4-trimethyl- | aromatics | aromatics | 0.63 | 8.87 | | 32.50 | Atkinson and Arey,2003 | 0.06 | Chan et al., 2009 |
| Benzene, 1-ethyl-3-methyl- | aromatics | aromatics | 0.63 | 7.39 | | 18.60 | Atkinson and Arey,2003 | 0.10 | Chan et al., 2009 |
| o-Cymene | aromatics | aromatics | 0.63 | 5.49 | | 8.54 | AopWin | 0.06 | Benzene, 1,2,4-trimethyl- |
| 2-Methylphenylacetylene | aromatics | aromatics | 0.63 | 1.73 | Styrene | | | 0.06 | Benzene, 1,2,4-trimethyl- |
| Benzene, 1-methyl-2-propyl- | aromatics | aromatics | 0.63 | 5.49 | | 8.80 | AopWin | 0.06 | Benzene, 1,2,4-trimethyl- |
| aromatics-C4-surrogate | aromatics | aromatics | 0.63 | 2.36 | | 8.72 | AopWin | 0.10 | Benzene, propyl- |
| Benzene, 2,4-dimethyl-1-(1-methylethyl)- | aromatics | aromatics | 0.63 | 8.87 | Benzene, 1,2,4-trimethyl- | | | 0.10 | Benzene, propyl- |
| Benzene, hexyl- | aromatics | aromatics | 0.63 | 2.12 | Benzene, pentyl- | | | 0.10 | Benzene, propyl- |
| Benzene, (1-methylnonyl)- | aromatics | aromatics | 0.97 | 7.39 | Benzene, 1-ethyl-3-methyl- | | | 0.10 | Benzene, propyl- |
| 1H-Indene, 2,3-dihydro-1,1,3-trimethyl-3-phenyl- | aromatics | aromatics | 0.97 | 1.73 | Styrene | | | 0.10 | Benzene, propyl- |
| 2,4-Diphenyl-4-methyl-1-pentene | aromatics | aromatics | 0.97 | 2.12 | Benzene, pentyl- | | | 0.10 | Benzene, propyl- |
| Benzene, 1,1'-(1,1,2,2-tetramethyl-1,2-ethanediyl)bis- | aromatics | aromatics | 0.97 | 7.39 | Benzene, 1-ethyl-3-methyl- | | | 0.10 | Benzene, propyl- |
| 2,4-Diphenyl-4-methy | aromatics | aromatics | 0.97 | 2.12 | Benzene, pentyl- | | | 0.10 | Benzene, propyl- |

| Compound | Group | Category | | | | | | | |
|---|---|---|---|---|---|---|---|---|---|
| l-2(E)-pentene | | | | | | | | | |
| Benzene, 1,1'-(3,3-dimethyl-1-butenylidene)bis- | aromatics | aromatics | 0.97 | 7.39 | Benzene, 1-ethyl-3-methyl- | | | 0.10 | Benzene, propyl- |
| Benzene, propyl- | aromatics | aromatics | 0.88 | 2.03 | | 5.80 | Atkinson and Arey,2003 | 0.10 | Chan et al., 2009 |
| aromatics-C3 | aromatics | aromatics | 0.63 | 2.03 | | 5.80 | Atkinson and Arey,2003 | 0.10 | Chan et al., 2009 |
| aromatics-C4 | aromatics | aromatics | 0.63 | 2.36 | | 8.72 | AopWin | 0.10 | Benzene, propyl- |
| Benzene, pentyl- | aromatics | aromatics | 0.63 | 2.12 | | 10.14 | AopWin | 0.10 | Benzene, propyl- |
| Benzene, 1-methyl-3-propyl- | aromatics | aromatics | 0.63 | 7.10 | | 15.25 | AopWin | 0.10 | Benzene, propyl- |
| Indane | aromatics | aromatics | 0.63 | 3.32 | | 19.00 | Atkinson and Arey,2003 | 0.08 | Gentner, 2012 |
| 1H-Indene, 2,3-dihydro-4-methyl- | aromatics | aromatics | 0.63 | | | | | 0.08 | Indane |
| Indane, 1-methyl- | aromatics | aromatics | 0.63 | 3.32 | Indane | | | 0.08 | Indane |
| Phenol, 2-chloro- | chlorides | chlorides | 0.95 | | | 9.87 | AopWin v1.92 | 0.22 | C10 |
| Bis(2-chloro-1-methylethyl) ether | chlorides | chlorides | 0.82 | | | | | 0.22 | C10 |
| Trichloroethylene | chlorides | chlorides | 0.82 | 0.64 | | 0.80 | AopWin | 0.06 | C8 |
| Tetrachloroethylene | chlorides | chlorides | 0.82 | 0.03 | | 0.21 | AopWin | 0.06 | C8 |
| Phenol, 4-chloro-3-methyl- | chlorides | chlorides | 0.96 | | | | | 0.38 | Phenol |
| N-Nitrosodimethylamine | amines | nitrogen-containing compounds | 0.76 | | | | | 0.06 | C8 |
| Cyclohexane, isocyanato- | CN | nitrogen-containing compounds | 0.92 | | | | | 0.22 | C10 |
| Nitric acid, pentyl ester | nitrates | nitrogen-containing | 0.93 | | | | | 0.14 | C9 |

| | | compounds | | | | | |
|---|---|---|---|---|---|---|---|
| **Decanenitrile** | nitriles | nitrogen-containing compounds | 0.99 | 8.74 | AopWin v1.92 | 0.03 | C13 |
| **Benzonitrile** | nitriles | nitrogen-containing compounds | 0.75 | 0.34 | AopWin | 0.22 | C10 |
| **o-Nitroaniline** | nitro | nitrogen-containing compounds | 0.89 | 13.45 | AopWin v1.92 | 0.05 | C14 |
| **Pentane, 1-nitro-** | nitro-alkanes | nitrogen-containing compounds | 0.92 | | | 0.14 | C9 |
| **Benzene, 2-methyl-1,3-dinitro-** | nitrophenols | nitrogen-containing compounds | 0.96 | 0.27 | AopWin v1.92 | 0.05 | C14 |
| **Benzene, 1-methyl-2,4-dinitro-** | nitrophenols | nitrogen-containing compounds | 0.96 | 0.27 | AopWin v1.92 | 0.08 | C15 |
| **Pyridine, 2-pentyl-** | pyridines | nitrogen-containing compounds | 0.97 | | | 0.02 | C12 |
| **Benzothiazole** | SN | nitrogen-containing compounds | 0.97 | | | 0.02 | C12 |
| **Cyclohexane, isothiocyanato-** | SN | nitrogen-containing compounds | 0.97 | | | 0.02 | C12 |
| **1,2-Benzisothiazole** | SN | nitrogen-containing compounds | 0.97 | | | 0.02 | C12 |

| | | taining compounds | | | | | | | |
|---|---|---|---|---|---|---|---|---|---|
| **Undecanoic acid** | acids | oxygenated compounds | 0.97 | | | 12.59 | AopWin v1.92 | 0.05 | C14 |
| **Tridecanoic acid** | acids | oxygenated compounds | 0.88 | | | 15.42 | AopWin v1.92 | 0.12 | C16 |
| **Acetic acid** | acids | oxygenated compounds | 0.32 | 0.68 | | 0.62 | AopWin | | |
| **Butanoic acid, 3-methyl-** | acids | oxygenated compounds | 0.92 | 4.23 | | 4.10 | AopWin | 0.06 | C8 |
| **Butanoic acid, 2-methyl-** | acids | oxygenated compounds | 0.92 | 4.23 | Butanoic acid, 3-methyl- | | | 0.06 | C8 |
| **Pentanoic acid** | acids | oxygenated compounds | 0.32 | | | 4.11 | AopWin | 0.14 | C9 |
| **Hexanoic acid** | acids | oxygenated compounds | 0.32 | | | 5.52 | AopWin v1.92 | 0.22 | C10 |
| **Heptanoic acid** | acids | oxygenated compounds | 0.81 | | | 6.94 | AopWin v1.92 | 0.33 | C11 |
| **Benzoic acid** | acids | oxygenated compounds | 0.32 | | | 1.24 | AopWin v1.92 | 0.02 | C12 |
| **Octanoic acid** | acids | oxygenated compounds | 0.32 | | | | | 0.02 | C12 |
| **Nonanoic acid** | acids | oxygenated compounds | 0.32 | | | 9.76 | AopWin v1.92 | 0.03 | C13 |
| **Phenylmaleic anhydride** | acids | oxygenated compounds | 0.88 | | | | | 0.08 | C15 |
| **2-Hexenal, (E)-** | aldehyde-enes | oxygenated compounds | 0.96 | | 2-Hexenal | | | 0.02 | pentanal |
| **Furfural** | aldehyde-e | oxygenated | 0.96 | | | 37.42 | AopWin | 0.02 | pentanal |

| | | | | | | | | | |
|---|---|---|---|---|---|---|---|---|---|
| | nes | compounds | | | | | | | |
| 2-Hexenal | aldehyde-enes | oxygenated compounds | 0.96 | | | 38.52 | AopWin | 0.02 | pentanal |
| 4-Heptenal, (Z)- | aldehyde-enes | oxygenated compounds | 0.96 | 2-Hexenal | | | | 0.02 | pentanal |
| 2-Heptenal, (Z)- | aldehyde-enes | oxygenated compounds | 0.96 | 2-Hexenal | | | | 0.02 | pentanal |
| 4-Oxohex-2-enal | aldehyde-enes | oxygenated compounds | 0.96 | 2-Hexenal | | | | 0.02 | pentanal |
| aldehyde-enes-trans-2-Dodecenal-surrogate | aldehyde-enes | oxygenated compounds | 0.96 | 2-Hexenal | | | | 0.02 | pentanal |
| 2-Heptenal, (E)- | aldehyde-enes | oxygenated compounds | 0.96 | 2-Hexenal | | | | 0.02 | pentanal |
| 2,4-Heptadienal | aldehyde-enes | oxygenated compounds | 0.98 | 2-Hexenal | | | | 0.02 | pentanal |
| 2,4-Heptadienal, (E,E)- | aldehyde-enes | oxygenated compounds | 0.98 | 2-Hexenal | | | | 0.02 | pentanal |
| 2-Octenal, (E)- | aldehyde-enes | oxygenated compounds | 0.98 | 2-Hexenal | | | | 0.02 | pentanal |
| 4-Nonenal, (E)- | aldehyde-enes | oxygenated compounds | 0.98 | 2-Hexenal | | | | 0.02 | pentanal |
| 2-Nonenal, (Z)- | aldehyde-enes | oxygenated compounds | 0.98 | 2-Hexenal | | | | 0.02 | pentanal |
| 2-Nonenal, (E)- | aldehyde-enes | oxygenated compounds | 0.98 | 2-Hexenal | | | | 0.02 | pentanal |
| 2,4-Nonadienal | aldehyde-enes | oxygenated compounds | 0.98 | 2-Hexenal | | | | 0.02 | pentanal |
| 4-Decenal, (E)- | aldehyde-enes | oxygenated compounds | 0.98 | 2-Hexenal | | | | 0.02 | pentanal |

| | | | | | | | |
|---|---|---|---|---|---|---|---|
| 2,4-Nonadienal, (E,E)- | aldehyde-enes | oxygenated compounds | 0.98 | 2-Hexenal | | 0.02 | pentanal |
| 2-Decenal, (Z)- | aldehyde-enes | oxygenated compounds | 0.98 | 2-Hexenal | | 0.02 | pentanal |
| (Z)-3-Phenylacrylaldehyde | aldehyde-enes | oxygenated compounds | 0.98 | | | 0.02 | pentanal |
| 2-Decenal, (E)- | aldehyde-enes | oxygenated compounds | 0.98 | 2-Hexenal | | 0.02 | pentanal |
| 2,4-Decadienal, (E,Z)- | aldehyde-enes | oxygenated compounds | 0.98 | 2-Hexenal | | 0.02 | pentanal |
| cis-Undec-4-enal | aldehyde-enes | oxygenated compounds | 0.98 | 2-Hexenal | | 0.02 | pentanal |
| 2,4-Decadienal, (E,E)- | aldehyde-enes | oxygenated compounds | 0.98 | 2-Hexenal | | 0.02 | pentanal |
| 2-Undecenal, E- | aldehyde-enes | oxygenated compounds | 0.98 | 2-Hexenal | | 0.02 | pentanal |
| 2,4-Decadienal | aldehyde-enes | oxygenated compounds | 0.98 | 2-Hexenal | | 0.02 | pentanal |
| 2-Undecenal | aldehyde-enes | oxygenated compounds | 0.98 | 2-Hexenal | | 0.02 | pentanal |
| 2,4-Dodecadienal | aldehyde-enes | oxygenated compounds | 0.98 | 2-Hexenal | | 0.02 | pentanal |
| 2-Dodecenal | aldehyde-enes | oxygenated compounds | 0.98 | 2-Hexenal | | 0.02 | pentanal |
| 7,11-Hexadecadienal | aldehyde-enes | oxygenated compounds | 0.98 | 2-Hexenal | | 0.02 | pentanal |
| Neophytadiene | aldehyde-enes | oxygenated compounds | 0.96 | | | 0.41 | alpha-Pinene |

| | | | | | | | | | | |
|---|---|---|---|---|---|---|---|---|---|---|
| **Pentanal** | aldehydes | oxygenated compounds | 0.96 | 4.35 | Hexanal | 28.00 | Atkinson and Arey,2003 | 0.02 | | Chan et al., 2009 |
| **2-Furanol, tetrahydro-** | aldehydes | oxygenated compounds | 0.96 | | | | | 0.02 | pentanal | |
| **Hexanal** | aldehydes | oxygenated compounds | 0.96 | 4.35 | | 30.00 | Atkinson and Arey,2003 | 0.02 | pentanal | |
| **Heptanal** | aldehydes | oxygenated compounds | 0.97 | 3.69 | | 30.00 | Atkinson and Arey,2003 | 0.02 | pentanal | |
| **Benzaldehyde** | aldehydes | oxygenated compounds | 0.96 | | | 12.00 | Atkinson and Arey,2003 | 0.38 | | Fang et al., 2017 |
| **Octanal** | aldehydes | oxygenated compounds | 0.97 | 3.16 | | 31.66 | AopWin v1.92 | 0.02 | pentanal | |
| **3-Cyclohexene-1-carb oxaldehyde, 1-methyl-** | aldehydes | oxygenated compounds | 0.98 | | | | | 0.02 | pentanal | |
| **Benzeneacetaldehyde** | aldehydes | oxygenated compounds | 0.98 | | Benzeneacetalde hyde | 26.31 | AopWin v1.92 | 0.38 | benzaldehyde | |
| **Nonanal** | aldehydes | oxygenated compounds | 0.98 | 3.16 | Octanal | 33.07 | AopWin v1.92 | 0.02 | pentanal | |
| **Decanal** | aldehydes | oxygenated compounds | 0.98 | 3.16 | Octanal | 34.48 | AopWin v1.92 | 0.02 | pentanal | |
| **2-Sec-Butylcyclohexa none** | aldehydes | oxygenated compounds | 0.98 | | | | | 0.02 | pentanal | |
| **4-Oxononanal** | aldehydes | oxygenated compounds | 0.98 | | | | | 0.02 | pentanal | |
| **Cyclohexanone, 2-butyl-** | aldehydes | oxygenated compounds | 0.98 | | | | | 0.02 | pentanal | |
| **Undecanal** | aldehydes | oxygenated compounds | 0.98 | 3.16 | Octanal | | | 0.02 | pentanal | |
| **Dodecanal** | aldehydes | oxygenated | 0.98 | 3.16 | Octanal | | | 0.02 | pentanal | |

| | | | | | | | | | | |
|---|---|---|---|---|---|---|---|---|---|---|
| | | compounds | | | | | | | | |
| **Tridecanal** | aldehydes | oxygenated compounds | 0.98 | 3.16 | Octanal | | | 0.02 | pentanal | |
| **1-Hexanol** | alkanols | oxygenated compounds | 0.96 | 2.69 | | 15.00 | Atkinson and Arey,2003 | 0.00 | 1-butanol | |
| **1-Heptanol** | alkanols | oxygenated compounds | 0.95 | 1.84 | | 14.00 | Atkinson and Arey,2003 | 0.05 | n-heptane | |
| **1-Decanol** | alkanols | oxygenated compounds | 0.97 | 1.43 | 1-Octanol | 15.37 | AopWin v1.92 | 0.50 | | Lucas B. Algrim,2019 |
| **1-Butanol** | alkanols | oxygenated compounds | 0.78 | 2.88 | | 8.50 | Atkinson and Arey,2003 | 0.00 | 1-butanol | Wu et al., 2017 |
| **1-Pentanol** | alkanols | oxygenated compounds | 0.78 | 2.83 | | 11.00 | Atkinson and Arey,2003 | 0.00 | 1-butanol | |
| **3,3-Dimethylbutane-2 -ol** | alkanols | oxygenated compounds | 0.78 | | | | | 0.05 | n-heptane | |
| **Cyclopentanol, 2-methyl-, trans-** | alkanols | oxygenated compounds | 0.78 | | | | | 0.05 | n-heptane | |
| **2-Heptanol** | alkanols | oxygenated compounds | 0.84 | 1.84 | 1-Heptanol | | | 0.05 | n-heptane | |
| **2-Octanol** | alkanols | oxygenated compounds | 0.80 | 1.43 | 1-Octanol | | | 0.06 | C8 | |
| **Cyclohexanol, 2,4-dimethyl-** | alkanols | oxygenated compounds | 0.80 | | | | | 0.06 | C8 | |
| **3,4-Dimethylcyclohex anol** | alkanols | oxygenated compounds | 0.80 | | | | | 0.06 | C8 | |
| **1-Octanol** | alkanols | oxygenated compounds | 0.99 | 1.43 | | 14.00 | Atkinson and Arey,2003 | 0.50 | 1-Decanol | |
| **1-Nonanol** | alkanols | oxygenated compounds | 0.97 | 1.43 | 1-Octanol | 13.96 | AopWin v1.92 | 0.50 | 1-Decanol | |

| Compound | Class | Category | | | | | | | | Source |
|---|---|---|---|---|---|---|---|---|---|---|
| 6-Undecanol | alkanols | oxygenated compounds | 0.65 | 1.43 | 1-Octanol | | | 0.10 | 5-Decanol | Lucas B. Algrim,2019 |
| 1-Undecanol | alkanols | oxygenated compounds | 0.99 | 1.43 | 1-Octanol | 16.78 | AopWin v1.92 | 0.50 | 1-Decanol | |
| 1-Octen-3-ol | alkanols | oxygenated compounds | 0.84 | | | | | 0.05 | 1-Octene | |
| 2-Octen-1-ol, (E)- | alkanols | oxygenated compounds | 0.80 | | | | | 0.05 | 1-Octene | |
| alkenols-1-Tridecanol-surrogate | alkanols | oxygenated compounds | 0.65 | | | | | 0.46 | alkenes-C13 | |
| 1,2-Heptanediol | di-ols | oxygenated compounds | 0.84 | | | | | 0.05 | n-heptane | |
| Benzene, 1-methoxy-4-(1-propenyl)-, (Z)- | esters | oxygenated compounds | 0.69 | | | | | 0.10 | Benzene, propyl- | |
| 2(3H)-Furanone, dihydro-3-methyl- | esters | oxygenated compounds | 0.93 | | | 2.72 | AopWin v1.92 | 0.14 | C9 | |
| 2(3H)-Furanone, dihydro-5-methyl- | esters | oxygenated compounds | 0.93 | | | | | 0.14 | C9 | |
| 2H-Pyran-2-one, tetrahydro-3-methyl- | esters | oxygenated compounds | 0.66 | | | | | 0.33 | C11 | |
| Methyl myristoleate | esters | oxygenated compounds | 0.99 | 0.44 | Hexadecanoic acid, methyl ester | | | 0.20 | C17 | |
| Benzoic acid, 2-ethylhexyl ester | esters | oxygenated compounds | 0.89 | 0.98 | | 11.54 | AopWin | 0.20 | C17 | |
| Methyl (Z)-10-pentadecenoate | esters | oxygenated compounds | 0.99 | 1.70 | 9-Hexadecenoic acid, methyl ester, (Z)- | | | 0.30 | C18 | |
| 9-Hexadecenoic acid, | esters | oxygenated compounds | 0.98 | 1.70 | | 71.89 | AopWin | 0.42 | C19 | |

| | | | | | | | | | |
|---|---|---|---|---|---|---|---|---|---|
| methyl ester, (Z)- | | compounds | | | | | | | |
| Methyl gamma linolenate | esters | oxygenated compounds | 0.98 | 2.32 | | 180.96 | AopWin | 0.56 | C20 |
| 9-Octadecenoic acid (Z)-, methyl ester | esters | oxygenated compounds | 0.98 | 1.54 | | 74.72 | AopWin | 0.77 | C21 |
| 9,12-Octadecadienoic acid (Z,Z)-, methyl ester | esters | oxygenated compounds | 0.98 | 1.84 | | 127.81 | AopWin | 0.77 | C21 |
| 9-Octadecenoic acid, methyl ester, (E)- | esters | oxygenated compounds | 0.99 | 1.54 | 9-Octadecenoic acid (Z)-, methyl ester | | | 0.77 | C21 |
| 5,8,11,14,17-Eicosapentaenoic acid, methyl ester, (all-Z)- | esters | oxygenated compounds | 0.95 | 1.84 | 9,12-Octadecadienoic acid (Z,Z)-, methyl ester | | | 0.96 | C22 |
| 5,8,11,14-Eicosatetraenoic acid, methyl ester, (all-Z)- | esters | oxygenated compounds | 0.98 | 1.84 | 9,12-Octadecadienoic acid (Z,Z)-, methyl ester | | | 1.08 | C23 |
| cis-11,14,17-Eicosatrienoic acid, methyl ester | esters | oxygenated compounds | 0.97 | 1.84 | 9,12-Octadecadienoic acid (Z,Z)-, methyl ester | | | 1.08 | C23 |
| 4,7,10,13,16,19-Docosahexaenoic acid, methyl ester, (all-Z)- | esters | oxygenated compounds | 0.95 | 1.84 | 9,12-Octadecadienoic acid (Z,Z)-, methyl ester | | | 1.14 | C24 |
| 13-Docosenoic acid, methyl ester | esters | oxygenated compounds | 0.98 | | | | | 1.14 | C24 |
| 15-Tetracosenoic acid, methyl ester, (Z)- | esters | oxygenated compounds | 0.90 | | | | | 1.14 | C24 |
| Ethyl Acetate | esters | oxygenated compounds | 0.93 | 0.63 | | 1.70 | AopWin | 0.06 | C8 |

| Name | Class | Type | V1 | V2 | Reference | Value | Method | V3 | Code |
|---|---|---|---|---|---|---|---|---|---|
| **Acetic acid, butyl ester** | esters | oxygenated compounds | 0.93 | 0.83 | | 4.61 | AopWin | 0.06 | C8 |
| **Formic acid, pentyl ester** | esters | oxygenated compounds | 0.93 | 0.83 | Acetic acid, butyl ester | | | 0.06 | C8 |
| **Acetic acid, hexyl ester** | esters | oxygenated compounds | 0.66 | 0.83 | Acetic acid, butyl ester | | | 0.22 | C10 |
| **n-Caproic acid vinyl ester** | esters | oxygenated compounds | 0.66 | 0.83 | Acetic acid, butyl ester | | | 0.22 | C10 |
| **2(3H)-Furanone, 5-butyldihydro-** | esters | oxygenated compounds | 0.69 | | | | | 0.02 | C12 |
| **Hexanoic acid, pentyl ester** | esters | oxygenated compounds | 0.69 | 0.44 | Hexadecanoic acid, methyl ester | | | 0.03 | C13 |
| **Benzoic acid, 1-methylpropyl ester** | esters | oxygenated compounds | 0.69 | 0.98 | Benzoic acid, 2-ethylhexyl ester | | | 0.03 | C13 |
| **Benzoic acid, pentyl ester** | esters | oxygenated compounds | 0.95 | 0.98 | Benzoic acid, 2-ethylhexyl ester | | | 0.08 | C15 |
| **Hexadecanoic acid, methyl ester** | esters | oxygenated compounds | 0.97 | 0.44 | | 18.85 | AopWin | 0.42 | C19 |
| **1-Propene-1,2,3-tricarboxylic acid, tributyl ester** | esters | oxygenated compounds | 0.99 | | | | | 0.77 | C21 |
| **n-Amyl ether** | ethers | oxygenated compounds | 0.90 | 2.15 | | 27.52 | AopWin | 0.33 | C11 |
| **Butyrolactone** | furanones | oxygenated compounds | 0.93 | 0.96 | | 2.31 | AopWin | 0.14 | C9 |
| **4-Methyl-5H-furan-2-one** | furanones | oxygenated compounds | 0.93 | | | | | 0.22 | C10 |
| **2(3H)-Furanone, 5-ethyldihydro-** | furanones | oxygenated compounds | 0.93 | | | 5.45 | AopWin v1.92 | 0.22 | C10 |

| Compound | Subclass | Class | | | | | | | | |
|---|---|---|---|---|---|---|---|---|---|---|
| 2(5H)-Furanone, 5-(1-methylethyl)- | furanones | oxygenated compounds | 0.66 | | | | | 0.33 | C11 | |
| 2(3H)-Furanone, dihydro-5-propyl- | furanones | oxygenated compounds | 0.66 | | | | | 0.33 | C11 | |
| 2(3H)-Furanone, dihydro-5-pentyl- | furanones | oxygenated compounds | 0.69 | | | | | 0.03 | C13 | |
| 3-Furanmethanol | furans | oxygenated compounds | 0.78 | | | | | 0.06 | C8 | |
| Furan, 2-pentyl- | furans | oxygenated compounds | 0.84 | | | | | 0.22 | C10 | |
| 2-N-Octylfuran | furans | oxygenated compounds | 0.65 | | | | | 0.03 | C13 | |
| 1-Octen-3-one | ketone-enes | oxygenated compounds | 0.58 | 1.40 | 2-Octanone | | | 0.05 | 1-Octene | |
| trans-3-Nonen-2-one | ketone-enes | oxygenated compounds | 0.58 | | | | | 0.15 | 1-Nonene | |
| 2-Hexanone | ketones | oxygenated compounds | 0.96 | 3.14 | | 9.10 | Atkinson and Arey,2003 | 0.06 | C8 | Lucas B. Algrim,2016 |
| Cyclopentanone, 2-methyl- | ketones | oxygenated compounds | 0.96 | | | | | 0.06 | C8 | |
| 2-Heptanone | ketones | oxygenated compounds | 0.96 | 2.36 | | 11.00 | Atkinson and Arey,2003 | 0.14 | C9 | |
| 3-Ethylcyclopentanone | ketones | oxygenated compounds | 0.96 | 2.36 | 2-Heptanone | | | 0.14 | C9 | |
| 2-Octanone | ketones | oxygenated compounds | 0.96 | 1.40 | | 11.00 | Atkinson and Arey,2003 | 0.22 | C10 | |
| Acetophenone | ketones | oxygenated compounds | 0.96 | | | 1.88 | AopWin v1.92 | 0.38 | benzaldehyde | |
| Cyclopentanone, | ketones | oxygenated | 0.96 | | | | | 0.33 | C11 | |

| Name | | | | | | | | | |
|---|---|---|---|---|---|---|---|---|---|
| 3-butyl-1-Propanone, 1-phenyl- | ketones | oxygenated compounds | 0.96 | | | | | 0.38 | benzaldehyde |
| 6-Dodecanone | ketones | oxygenated compounds | 0.96 | 1.40 | 2-Octanone | | | 0.42 | Lucas B. Algrim,2016 |
| 1-Hexanone, 1-phenyl- | ketones | oxygenated compounds | 0.96 | | | | | 0.38 | benzaldehyde |
| 2-Pentadecanone | ketones | oxygenated compounds | 0.96 | 1.40 | 2-Octanone | | | 0.20 | C17 |
| 6-(p-Tolyl)-2-methyl-2-heptenol, trans- | oxgenated-tri-isoprenes | oxygenated compounds | 0.98 | | | | | 0.12 | C16 |
| oxiranes-surrogate-Oxirane, decyl- | oxiranes | oxygenated compounds | 0.98 | | | | | 0.33 | C11 |
| oxo-aldehyde-enes | oxo-aldehyde-enes | oxygenated compounds | 0.98 | | | | | 0.03 | C13 |
| cis-4,5-Epoxy-(E)-2-decenal | oxo-aldehyde-enes | oxygenated compounds | 0.98 | | | | | 0.03 | C13 |
| cis-2,3-Epoxyoctane | oxygenated-alkanes | oxygenated compounds | 0.98 | | | | | 0.14 | C9 |
| 3-Hydroxy-3-phenylbutan-2-one | oxygenated-aromatics | oxygenated compounds | 0.96 | | | | | 0.38 | Phenol |
| oxygenated-aromatics | oxygenated-aromatics | oxygenated compounds | 0.96 | | | | | 0.38 | Phenol |
| Estragole | oxygenated-aromatics | oxygenated compounds | 0.96 | | | 54.26 | AopWin | 0.38 | Phenol |
| 1,2-Benzenedicarboxylic acid | oxygenated-aromatics | oxygenated compounds | 0.96 | | | | | 0.38 | Phenol |
| Benzeneacetic acid, | oxygenated | oxygenated | 0.96 | | | | | 0.38 | Phenol |

| | | | | | | | | | |
|---|---|---|---|---|---|---|---|---|---|
| methyl ester | -aromatics | compounds | | | | | | | |
| 2,6-Di-tert-butyl-4-hydroxy-4-methylcyclohexa-2,5-dien-1-one | oxygenated-aromatics | oxygenated compounds | 0.96 | | | | | 0.38 | Phenol |
| o-Hydroxybiphenyl | oxygenated-aromatics | oxygenated compounds | 0.96 | | | | | 0.38 | Phenol |
| Benzophenone | oxygenated-aromatics | oxygenated compounds | 0.96 | | | 3.55 | AopWin v1.92 | 0.38 | Phenol |
| Xanthoxylin | oxygenated-aromatics | oxygenated compounds | 0.96 | | | | | 0.38 | Phenol |
| Ethanone, 1,2-diphenyl- | oxygenated-aromatics | oxygenated compounds | 0.96 | | | 7.32 | AopWin v1.92 | 0.38 | Phenol |
| 3,5-di-tert-Butyl-4-hydroxybenzaldehyde | oxygenated-aromatics | oxygenated compounds | 0.96 | | | | | 0.38 | Phenol |
| 1,7-Octadien-3-ol, 2,6-dimethyl- | oxygenated-bi-isoprenes | oxygenated compounds | 0.80 | | | | | 0.41 | alpha-Pinene |
| oxygenated-bi-isoprenes | oxygenated-bi-isoprenes | oxygenated compounds | 0.80 | | | | | 0.41 | alpha-Pinene |
| 8-Oxabicyclo[5.1.0]octane | oxygenated-cycloalkanes | oxygenated compounds | 0.78 | | | | | 0.41 | alpha-Pinene |
| Cyclohexanecarboxaldehyde | oxygenated-cycloalkanes | oxygenated compounds | 0.78 | | | | | 0.41 | alpha-Pinene |
| Eucalyptol | oxygenated-di-isoprenes | oxygenated compounds | 0.80 | 5.43 | Linalool | | | 0.41 | alpha-Pinene |

| | | | | | | | | |
|---|---|---|---|---|---|---|---|---|
| **oxygenated-di-isopre nes** | oxygenated -di-isopren es | oxygenated compounds | 0.80 | | | | 0.41 | alpha-Pinene |
| **Linalool** | oxygenated -di-isopren es | oxygenated compounds | 0.80 | 5.43 | | 119.6 4 | AopWin | 0.41 | alpha-Pinene |
| **3-Cyclohexen-1-ol, 4-methyl-1-(1-methyl ethyl)-, (R)-** | oxygenated -di-isopren es | oxygenated compounds | 1.00 | | | | 0.41 | alpha-Pinene |
| **3-Cyclohexene-1-met hanol, alpha,alpha,4-trimeth yl-, propanoate** | oxygenated -di-isopren es | oxygenated compounds | 1.00 | | | | 0.41 | alpha-Pinene |
| **2-Cyclohexen-1-one, 3-methyl-6-(1-methyl ethyl)-** | oxygenated -di-isopren es | oxygenated compounds | 1.00 | | | | 0.41 | alpha-Pinene |
| **2,4-Pentadien-1-ol, 3-pentyl-, (2Z)-** | oxygenated -di-isopren es | oxygenated compounds | 1.00 | | | | 0.41 | alpha-Pinene |
| **Linalyl acetate** | oxygenated -di-isopren es | oxygenated compounds | 1.00 | | | | 0.41 | alpha-Pinene |
| **2H-1b,4-Ethanopenta leno[1,2-b]oxirene, hexahydro-, (1a-alpha-,1b-bta-,4-b ta-,4a-alpha-,5a-alpha -)-** | oxygenated -di-isopren es | oxygenated compounds | 0.65 | | | | 0.41 | alpha-Pinene |
| **alpha-Terpinyl** | oxygenated | oxygenated | 0.65 | | | | 0.41 | alpha-Pinene |

| | | | | | | | | | | |
|---|---|---|---|---|---|---|---|---|---|---|
| **acetate** | -di-isoprenes | compounds | | | | | | | | |
| **1-Penten-3-ol** | oxygenated-isoprenes | oxygenated compounds | 0.78 | | | | | 0.41 | alpha-Pinene | |
| **Phenol** | phenols | oxygenated compounds | 0.96 | 2.76 | | 33.47 | AopWin v1.92 | 0.38 | | Fang et al., 2017 |
| **p-Cresol** | phenols | oxygenated compounds | 0.95 | 2.40 | | 41.13 | AopWin v1.92 | 0.38 | Phenol | |
| **Phenol, 2,4-dimethyl-** | phenols | oxygenated compounds | 0.98 | 2.12 | | 50.49 | AopWin v1.92 | 0.38 | Phenol | |
| **2H-Pyran-2-one, tetrahydro-** | pyranones | oxygenated compounds | 0.66 | | | | | 0.22 | C10 | |
| **Furan, 2-butyltetrahydro-** | tetrahydro-furans | oxygenated compounds | 0.78 | 2.13 | | 23.56 | AopWin | 0.22 | C10 | |
| **Naphthalene, 2-methyl-** | PAHs | PAHs | 0.93 | 3.06 | | 48.60 | Phousongphouang and Arey, 2002 | 0.38 | | Chan et al., 2009 |
| **Acenaphthylene** | PAHs | PAHs | 0.99 | 3.34 | Naphthalene | 75.49 | AopWin v1.92 | 0.03 | | Fang et al., 2017 |
| **Anthracene** | PAHs | PAHs | 1.00 | 3.34 | Naphthalene | 40.00 | AopWin v1.92 | 0.49 | | Gentner, 2012 |
| **Naphthalene** | PAHs | PAHs | 0.98 | 3.34 | | 23.00 | Atkinson and Arey,2003 | 0.26 | | Chan et al., 2009 |
| **Naphthalene, 1-methyl-** | PAHs | PAHs | 0.93 | 3.06 | | 40.90 | Phousongphouang and Arey, 2002 | 0.33 | | Chan et al., 2009 |
| **Phenanthrene** | PAHs | PAHs | 0.99 | 3.34 | Naphthalene | 13.00 | AopWin v1.92 | 0.49 | | Gentner, 2012 |
| **Silane, diethoxydiphenyl-** | siloxanes | siloxanes | 0.97 | | | | | 0.10 | Benzene, propyl- | |
| **UCM3** | UCMs | UCMs | 0.92 | 0.68 | C10 | | | 0.22 | C10 | |
| **UCMs** | UCMs | UCMs | 0.90 | 0.61 | C11 | | | 0.33 | C11 | |
| **UCM6** | UCMs | UCMs | 0.90 | 0.61 | C11 | | | 0.33 | C11 | |
| **UCM5** | UCMs | UCMs | 0.99 | 0.55 | C12 | | | 0.02 | C12 | |
| **UCM1** | UCMs | UCMs | 0.94 | 0.53 | C13 | | | 0.03 | C13 | |

| | | | | | | | | |
|---|---|---|---|---|---|---|---|---|
| **UCM2** | UCMs | UCMs | 0.94 | 0.53 | C13 | | 0.03 | C13 |
| **UCM4** | UCMs | UCMs | 0.94 | 0.53 | C13 | | 0.03 | C13 |
| **UCM7** | UCMs | UCMs | 0.93 | 0.51 | C14 | | 0.05 | C14 |
| **UCM8** | UCMs | UCMs | 0.93 | 0.51 | C14 | | 0.05 | C14 |
| **UCM9** | UCMs | UCMs | 0.93 | 0.51 | C14 | | 0.05 | C14 |
| **2,5-Cyclohexadiene-1, 4-dione, 2,6-bis(1,1-dimethylet hyl)-** | UCMs | UCMs | 0.93 | | | | 0.05 | C14 |

**Reference:**

[revised manuscript text omitted]

---

## Author Comment (AC3)

We thank the reviewers for their careful review of the manuscript. The comments greatly improved our manuscript. We revised our manuscript according to the reviewers' comments and suggestions. Overall, we have changed the mass concentration ($\mu g\ m^{-3}$) to the emission rate ($\mu g\ min^{-1}$) to avoid the influence of cooking time and sampling time according to the comments of the referees. We add more details to the volatility distributions of cooking emissions. We also added more comparisons with different studies. Following are our responses to the comments.

**Response to referee #3:**

This manuscript investigates the impact of cooking style and oil on the emissions from traditional Chinese cooking. A significant number of chemical species including aromatics, alkanes, oxygenated compounds, and PAHs have been detected. The authors observed that in addition to VOC species, S/IVOCs made up an important fraction of cooking emissions and SOA precursors. In general, dishes cooked by stir-frying and deep-frying styles emit more pollutants than relatively mild cooking styles. A volatility-polarity distribution framework of cooking emissions has been developed. Unlike the emissions that showed great variation, the volatility-polarity distribution of different cooking styles was similar. PLS-DA and MPCA analyses revealed that cooking oil was a critical influencing factor in the 2D distribution. Overall, this is a comprehensive study investigating the relationship among cooking emissions, cooking styles, and cooking materials. The manuscript is well written, and the results are valuable to the literature. I would like to recommend its publication in Atmospheric Chemistry and Physics, subject to minor revisions.

Thank you for your comments. The valuable suggestions addressed have greatly improved our manuscript. Following are our point-to-point responses to the comments.

1.   Table S1: In regard to oil temperature, how was oil temperature measured and monitored? Was oil temperature controlled and maintained the same during the cooking? There seems to be a positive relationship between oil temperature (Table S1) and emissions (Figure S3). Have the authors tried to cook the dishes at the same oil temperature and compare the emission results?

Thank you for your comment. The oil temperature was measured by a thermometer placed in the oil.

The thermometer was removed from the oil before placing the cooking materials. As a result, the

*initial* temperature of the oil was maintained the same for each dish. Dishes cooked at the same oil temperature were not conducted in this work. Further investigation will be carried on to illustrate the relationship between oil temperature and cooking emissions.

We revised Table S1 as follows.

**Table S1.** Details of cooking procedures.

| Domestic cooking | Material | Oil temperature [#] |
|---|---|---|
| Fried chicken | 170 g chicken, 500 mL oil (corn, peanut, soybean, or sunflower oil), a few condiments | 145 ~ 150 ℃ |
| Kung Pao chicken | 150 g chicken, 50 g peanut, 40 mL corn oil, a few condiments | Not stable |
| Pan-fried tofu | 500 g tofu, 200 mL corn oil, a few condiments | 100 ~ 110 ℃ |
| Stir-fried cabbage | 300 g chicken, 40 mL corn oil, a few condiments | 95 ~ 105 ℃ |

[#] The oil temperature was measured by a thermometer placed in the oil. The thermometer was removed from the oil before placing the cooking materials. The temperatures listed in Table S1 were initial cooking temperatures and were maintained the same for each dish.

2.      Line 117: What's the dimension of the Tenax TA tube? A flow rate of 0.5 L min-1 was used in this study. Do you have any idea what were the collection efficiencies of chemical species with different volatility under this flow rate condition? How long was the sampling? What about the breakthrough of Tenax TA tubes?

Thank you for your comment. The Tenax TA tube is Gerstel 6 mm 97 OD,  4.5  mm  ID  glass tube filled with  ~290  mg  Tenax  TA. A Tenax TA breakthrough experiment was conducted by introducing pure nitrogen gas ($N_2$) with a flow of 0.5 L min$^{-1}$ to the desorption tube with pre-added standard chemicals (Figure S2). No significant breakthrough was observed within 24 h (<3% of TIC).

The sampling time in this work is 15 ~ 30 min (0.5 L min$^{-1}$) which is much less than 24h. The chemical species quantified in this work was stable on Tenax TA tubes even after 24h of N$_2$ flowing. We have revised the manuscript accordingly.

Cooking fumes were sampled directly without dilution. After collecting particles on quartz filters, gas-phase organics were sampled by pre-conditioned Tenax TA tubes (Gerstel 6 mm 97 OD, 4.5 mm ID glass tube filled with ~290 mg Tenax TA) with a flow of 0.5 L min$^{-1}$. The removal of particles on the quartz filter in front of the Tenax TA tubes affects the S/IVOC measurements, causing positive and negative artifacts. Some of the gaseous SVOCs could be lost to sorption onto filters, and some particle-phase SVOCs could evaporate off the filter. The emission pattern of the particulate organics diverged from gas-phase organics, and a small overlap of species is identified. Aromatics, aldehydes, and short-chain acids mainly occurred in the gas-phase. For instance, the detection of short-chain olefinic aldehydes in the gas-phase was 40 times that of the particle-phase aldehydes. The artifacts of particulates on gas-phase aromatics and oxygenated compounds could be less than 5%. A typical system blank chromatogram is displayed in Figure S1. A daily blank sampling of the air in the kitchen ventilator was conducted before cooking and was subtracted in the quantification procedure. All samples were frozen at -20°C before analyzing. A Tenax TA breakthrough experiment was conducted by introducing pure nitrogen gas (N$_2$) with a flow of 0.5 L min$^{-1}$ to the desorption tube with pre-added standard chemicals (Figure S2). No significant breakthrough was observed within 24 h (<3% of TIC). The sampling time in this work is 15 ~ 30 min (0.5 L min$^{-1}$) which is much less than 24h.

[Figure]

Figure S2. The chromatograms of standard chemicals after 6h(brown), 24h(blue), 48h (red), and

72h (blue) of flowing by pure nitrogen gas. The flow of nitrogen gas is set to be the same as the sampling flow (0.5 L min⁻¹). No significant breakthrough was observed within 24 h (<3%).

3.    Lines 120-131: Chemical analysis using TD may have the following concerns (taking SVOCs as examples):

a)   Some of the SVOCs are of relatively low volatility. A TD temperature of 280 °C may not be sufficient to thermally released all the SVOCs in a short period of time.

b)   SVOCs such as acids may get decomposed during the TD processes.

c)   The decomposition of SVOCs may produce small molecules that can be mistakenly identified as VOCs.

Both items a and b lead to underestimations of SVOCs. Item c may result in an overestimation of

VOCs. In regard to these concerns, how long was the TD process in this study? Have the authors quantified the desorption efficiency of SVOC standards?

Thank you for your comment. The programming of the TD process was ramped 30℃ to 280℃

(60℃/min) and then retained at 280℃ for 10 min (Table S2). The total thermal desorption time was

14 min. 280℃ was chosen for thermal desorption temperature due to the less bleeding of Tenax TA

compared with 300℃. The linearities of undecanoic acid (C11-acid), C31, and C32 were 0.97, 0.99, and 0.99 (Table S5). The good linearity of SVOC compounds under different concentration levels showed a good desorption efficiency of SVOCs. Furthermore, the deportation of SVOC occurred in both *standards* and *samples*, making the quantification face less uncertainty. Though the direct desorption efficiency of SVOC is not quantified, we add more uncertainty discussions to the implication part of the manuscript as follows.

We still need to stress that although GC×GC is utilized, UCMs still occur sharing a proportion of 5% of the total response in this work. Acids and aldehydes tail in the second column and cause uncertainties in the quantification procedure. Meanwhile, TD-GC×GC-MS does not comprehensively measure all compounds. Acids can decompose during thermal desorption if no derivatization was performed. Meanwhile, the decomposition of SVOC compounds could produce small molecules in the VOC or IVOC range. The TD process could introduce underestimation for SVOC compounds while causing overestimations of VOC and IVOC species. Highly polar compounds do not elute from the GC column. This may lead to biases in estimating volatility and polarity distributions. Comparisons between GC×GC and chemical ionization mass spectrometers (CIMS) should be further implemented to give a full glimpse of cooking organic compounds.

4. Line 126: The authors mentioned that the chromatogram was cut into different volatility bins (B9 to B31 with a decrease in volatility). However, Figure 2 and Table S3 start from "B8_before". Please clarify.

Please add a sentence in the text defining the volatility of each bin (e.g., B8). Please also add a sentence in the text defining the polarity of each bin (e.g., P1). In this way, other studies can compare their results to this study when the volatility-polarity distribution framework is used.

Thank you for your comment. We have changed the statement of B9 to B8 as the 1D bins started with B8_before. We add instances of C12 and benzophenone to the main text to further illustrate the 2D binning method.

The total chromatogram was cut into volatility bins (B8 to B31 with a decrease in volatility) following the pipeline of previous studies (Tang et al., 2021; Zhao et al., 2014, 2017, 2018), while it was cut into slices by an increase of 0.5 s in the second retention time (called 2D bins, from P1 to P12 with an increase of polarity). For instance, C12 lies in B12 (saturated vapor concentration ~ $10^6$

µg m$^{-3}$, IVOC range) and P2 bins (low polarity). Benzophenone lies in B16 (saturated vapor concentration ~ $10^5$ µg m$^{-3}$, IVOC range) and P6 bins (medium to high polarity). A two-dimensional panel was developed in this way to investigate the emission of contaminants from aspects of their volatility and polarity properties (Song et al., 2022).

5.  Equation 2: SOA yield of VOC can increase with increasing particle loading (Odum et al.,

ES&T, 1996). Were the values of SOA yields used herein the maximum SOA yields? Please clarify.

Thank you for your comment. The SOA yields utilized in this work are under high $NO_x$ conditions which are underestimation of SOA due to the lower yields compared to low $NO_x$ conditions. We have revised the manuscript as follows.

SOA (µg min$^{-1}$) was estimated by the following equation, where $[HC_i]$ is the emission rate of precursor $i$ (µg min$^{-1}$) with OH reaction rate of $k_{OH,i}$, (cm$^3$ molecules$^{-1}$ s$^{-1}$) and SOA yield of $Y_i$

(Table S3). The SOA yields of precursors were from literature (Algrim and Ziemann, 2016, 2019;

Chan et al., 2009, 2010; Harvey and Petrucci, 2015; Li et al., 2016; Liu et al., 2018; Loza et al., 2014;

Matsunaga et al., 2009; McDonald et al., 2018; Shah et al., 2020; Tkacik et al., 2012; Wu et al., 2017)

or surrogates from $n$-alkanes in the same volatility bins (Zhao et al., 2014, 2017). The SOA yields utilized in this work are under high $NO_x$ conditions which are underestimation of SOA due to the lower yields compared to low $NO_x$ conditions. $[OH] \times \Delta t$ is the OH exposure and was set to be

14.4 $\times 10^{10}$ molecules cm$^{-3}$ s (~ 1.1 days in OH concentration of 1.5 $\times 10^6$ molecules cm$^{-3}$) in order to keep pace with our previous work (Zhang et al., 2021b; Zhu et al., 2021).

$$SOA = \sum [HC_i] \times (1 - e^{-k_{OH,i} \times [OH] \times \Delta t}) \times Y_i \quad (3)$$

6.  Lines 220-222: The authors mentioned that "an enhancement of ozone formation contribution and a decrease of SOA formation contribution were observed". The sentence is confusing. In regard to "enhancement" and "decrease", what were you comparing? Different types of VOCs, or VOCs vs.

S/IVOCs, or VOC emissions from different cooking styles?

Thank you for your comment. We compared the contribution to the mass proportion of VOCs in ERs.

We have revised the manuscript as follows.

Although chemicals in the VOC range dominated ozone and SOA formation, an increase in ozone formation contribution and a decrease in SOA formation contribution compared with the mass proportion of VOCs in ERs were observed. VOCs contributed 90.3% - 99.8% of the ozone estimation, and 68.0% - 89.8% of the total SOA estimation, compared with 81.4% - 95.6% in ERs. S/IVOCs explained 10.2% - 32.0% of the SOA estimation.

7.    Lines 236-237: The authors mentioned that "the emission patterns diverged from heated oil fumes as heated sunflower oil and peanut oil emitted more organics". It seems that this statement conflicts with the results shown in Figure S7 (dishes cooked by sunflower oil had the lowest emission).

Thank you for your comment. We have revised the manuscript as follows.

Chicken fried with corn oil emitted the most abundant gaseous contaminants. The emission patterns in this work diverged from heated oil fumes (Liu et al., 2018) as in their work heated sunflower oil and peanut oil emitted more organics.

[Figure]

Figure 4. Emission rate (ER), ozone formation potential (OFP), and secondary organic aerosol (SOA)

estimation from emissions of fried chicken cooked with corn, peanut, soybean, and sunflower oils.

The unit of the *y*-axis is μg min$^{-1}$.

8. Lines 265-266: "In contrast, the volatility-polarity distributions of dishes did not vary much when corn oil was used for cooking". Please add a reference to Figure 2.

Thank you for your comment. We have deleted this statement. The revised manuscript is shown as follows.

Although pollutants were dominated by aromatics, alkanes, and oxygenated compounds with volatility bins of B9 to B12 (VOC-IVOC range, saturated vapor concentration $> 10^6$ μg m$^{-3}$) and polarity bins of P1 to P5 (low to medium polarity), significant diversities of volatility-polarity distributions were observed (Figure S9). The chemical compositions in each volatility bin were also distinct (Figure S11). IVOCs accounted for as much as 22.8% and 23.7% of the total ERs when peanut and sunflower oils were utilized for frying (Kostik et al., 2013; Ryan et al., 2008). The peanut oil was much more abundant in oleic acid (41.5%), while the proportion of linoleic acid in sunflower is 36.6% (Figure S10). The proportion of unsaturated acids in peanut and sunflower oils is higher than that of other oils.

9.    Line 278: SOA production or reduction?

Thank you for your comment. We revised the manuscript as follows.

Despite the importance of aldehydes revealed in previous studies (Klein et al., 2016; Liu et al., 2018), our results demonstrated that alkanes, pinenes, and short-chain acids are also key precursors in cooking SOA **production** (Huang et al., 2020).

10.    Lines 294-295: What do you mean by "physical reactions (evaporation)"? Evaporation of what?

Thank you for your comment. We revised the manuscript as follows.

The PLS-DA result showed that cooking emissions diverged from oils (Figure 5 (c)), indicating that the physical reactions (**evaporation of edible oils**) were not the main reactions during the cooking procedure.

11.    Lines 295-296: "MPCA results showed the chromatogram similarities (positive loading) of oils and emissions." Please add a reference to Figure 3d. What is the color bar of Figure 3d?

Thank you for your comment. We add a reference to Figure 5d. The color bar in Figure 5(d) is the positive loading of pixels. We revised the manuscript as follows.

MPCA results showed the chromatogram similarities (positive loading) of oils and emissions (**Figure 5(d)**).

[Figure]

Figure 5. PLS-DA classification results in setting the cooking style (a) or oil (b) as grouping variables. When oil was set as the grouping variable, the separation of groups was much better than setting the dish as the grouping variable. The PLS-DA comparison result of cooking emissions and oils is displayed in (c), indicating that the cooking fume is not just the evaporation of oil itself.

Positive loadings of oil and cooking fume chromatograms (d) demonstrated the key components contributing to the similarities of samples. **The color bar in (d) is the positive loading of pixels.**

Technical comments:

1.    Line 167: duplicate word "form"

Thank you for your comment. We revised the manuscript as follows.

Chromatograms were imported from the network common data form (netCDF).

2.    Line 174: Change "results" to "result"

Thank you for your comment. We revised the manuscript as follows.

PLS-DA is a supervised method for the classification of grouped data. The main influencing factor could be apportioned if one separation **result** of PLS-DA is much better than the other.

3.    Line 313: Change "gas-phase" to "gas phase"

Thank you for your comment. We revised the manuscript as follows.

[revised manuscript text omitted]

---

## Author Response (AR2)

We thank the reviewer for his careful review of the manuscript. The comments greatly improved our manuscript. We revised our manuscript according to the reviewer's comments and suggestions. We did another breakthrough experiment by sampling two Tenax TA tubes in series.

**Response to referee #3:**

I do appreciate the time and the effort that the authors put into both revising the manuscript and replying to the comments from myself and other reviewers. The authors have sufficiently addressed most of my comments. In particular, the additional information presented in the new Figures S5, S10, and S11 is helpful.

We thank the reviewer for his careful review of the manuscript. Here is our point-to-point response.

I only have some concerns regarding Figure S2 about the breakthrough of the adsorbent tubes:

1. There are two "blue" figures referenced in the caption of Figure S2. One should be "black". Please correct the error.

Figure S2 has been replaced by chromatograms with two tubes sampled in series. Please refer to the updated Figure S2.

2. Comparing the bottom chromatogram to others ("black", "pink", and "blue"), I can actually see some differences (e.g., at 15-20 min and at 30-35 min). Can the authors show the differences between the normalized chromatograms of 24 h/48 h/72 h and that of 6 h?

Figure S2 has been replaced by chromatograms with two tubes sampled in series. Please refer to the updated Figure S2.

3. With regard to "breakthrough", I have a different understanding from the authors. Breakthrough of a specific analyte indicates the tube adsorption capacity (not strength) of the analyte, which varies from compound to compound and depends on sorbent-sorbate affinity.

Breakthrough can be normally examined by sampling with two adsorbent tubes (one sample tube + one backup tube) in series. After sampling, the two tubes should be analyzed in the same manner. If the mass of the analyte on the backup tube is at least a few percentages (e.g., >5%) of the mass on the front sampling tube, it indicates that the breakthrough of the analyte has occurred and the adsorption for the analyte on the front tube has reached saturation under the studied sampling

29     conditions (e.g., 15-30 min, 0.5 L min-1).

30     Thank you for your comments. We have deleted the previous Figure S2 and did another

31     breakthrough experiment.

32     The manuscript was revised as follows.

33         A Tenax TA breakthrough experiment was conducted by sampling two adsorbent tubes in series.

34     We sampled the first tube (sample tube) and the second tube (backup tube) simultaneously with a

35     sampling time of 24h. No breakthrough was observed after 24h sampling (Figure S2). The total

36     intensity of cooking emission chromatograms ($3.05 \times 10^9 - 14.17 \times 10^9$) falls in the range of the

37     sample tube ($9.84 \times 10^9$), which was much higher than the intensity of the backup tube ($2.12 \times 10^9$)

38     and the blank tube ($1.33 \times 10^9$, Figure S1). After subtracting the volume of the blank tube, the

39     volume of the backup tube is less than 10% of the sample tube, indicating the breakthrough effect of

40     the Tenax TA tubes could be neglected.

41

[Figure]

42

43     Figure S1. A typical chromatogram of system blank.

44

[Figure]

Figure S2. The chromatograms of the breakthrough experiment. (a) is the chromatogram of the sample tube, while (b) is the chromatogram of the backup tube. The sampling flow is set to be 0.5 L min$^{-1}$. No significant breakthrough was observed within 24 h (<1% for each compound). The total volume of cooking emission chromatograms ($3.05 \times 10^9 – 14.17 \times 10^9$) falls in the range of the sample tube ($9.84 \times 10^9$) and is much higher than the volume of the backup tube ($2.12 \times 10^9$). The volume of the backup tube ($2.12 \times 10^9$) is close to the volume of the blank tube ($1.33 \times 10^9$, Figure S1). After subtracting the volume of the blank tube, the volume of the backup tube is less than 10% of the sample tube, indicating the breakthrough effect of the Tenax TA tubes could be neglected.